# Bidirectional light-emitting diode as a visible light source driven by alternating current

Mikołaj Żak [1] ✉, Grzegorz Muziol [1], Marcin Siekacz [1], Artem Bercha [1], Mateusz Hajdel [1], Krzesimir Nowakowski-Szkudlarek [1], Artur Lachowski [1], Mikołaj Chlipała [1], Paweł Wolny [1], Henryk Turski [1] & Czesław Skierbiszewski [1]

Gallium nitride-based light-emitting diodes have revolutionized the lighting market by becoming the most energy-efficient light sources. However, the power grid, in example electricity delivery system, is built based on alternating current, which raises problems for directly driving light emitting diodes that require direct current to operate effectively. In this paper, we demonstrate a proof-of-concept device that addresses this fundamental issue – a gallium nitride-based bidirectional light-emitting diode. Its structure is symmetrical with respect to the active region, which, depending on the positive or negative bias, allows for the injection of either electrons or holes from each side. It is composed of two tunnel junctions that surround the active region. In this work, the optical and electrical properties of bidirectional light emitting diodes are investigated under direct and alternating current conditions. We find that the light is emitted in both directions of the supplied current, contrary to conventional light emitting diodes; hence, bidirectional light-emitting diodes can be considered a semiconductor light source powered directly with alternating current. In addition, we show that bidirectional light-emitting diodes can be stacked vertically to multiply the optical power achieved from a single device.

The III-nitride light-emitting diodes (III-N LEDs) coated with phosphors are currently the most efficient white light sources for general lighting[1–4]. The most efficient violet III-N LEDs reach approximately 80% of both external quantum efficiency and wall plug efficiency[5–8]. However, there is one crucial disadvantage of using LEDs for general lighting. The electric power in the electrical grid is distributed as alternating current (AC), whereas LEDs emit light only when the junction is forward biased. The AC is therefore converted into direct current (DC) to power LEDs. High-quality AC/DC converters are bulky, complex in design and always lead to inevitable power loss[9,10]. It was shown that the absence of an AC/DC converter saves up to 20% of electrical power under specific conditions. Great effort was made to fight this inconvenience[11–15]. Different approaches to obtain directly AC-driven light-emitting devices can be categorized into three groups.

The first group of AC LEDs is based on connecting a matrix of several LEDs both in series and in parallel. The LEDs can be connected in an antiparallel configuration or in a more sophisticated way to form a rectifier[11,12]. The advantage of this solution is that highly efficient III-N LEDs can be used. However, under AC conditions, only a portion of all LEDs used in the circuit emit light at the same time, which effectively reduces the surface power density received from the chips.

Another group of AC LEDs consists of devices based on organic emissive materials[13]. They are predominantly characterized by a symmetrical structure, in which an emitting layer is sandwiched between insulating dielectric layers and light is generated either by the hot-electron impact excitation mechanism or the radiative recombination of excitons. The disadvantage of this solution is a limited power

[1]Institute of High Pressure Physics Polish Academy of Sciences, Sokołowska 29/37, 01-142, Warsaw, Poland. ✉e-mail: mzak@unipress.waw.pl

efficiency, because achieving a proper balance between injected electrons and holes is difficult. In addition, they require a high operating voltage.

The last group of AC-driven devices involves specially designed structures based on III-nitrides, in which the phenomenon of carrier tunneling takes place. The first example of this group is a device presented by Hartensveld et al.[14]. The structure contains an InGaN quantum well (QW) and a thin $Al_2O_3$ barrier on top of it. There were no p-type layers; thus, the holes were injected into the QW by tunneling through the $Al_2O_3$ barrier. The authors have demonstrated a planar device with two identical $NiO/Al_2O_3$ contacts, which were able to inject holes or electrons into the QWs depending on the bias condition. The second example is a device presented by Sadaf et al.[15]. They have shown that the integration of two oppositely oriented III-N LEDs can be realized directly during the epitaxy process by using the selective area growth method. First, a conventional p-up LED was grown on half of the surface of a silicon wafer. Next, on the second half of the wafer, the growth was initiated with a tunnel junction (TJ), and then a p-down LED structure was grown. In this way, in the TJ-based p-down LED, the order of p-type and n-type in the LED was opposite, and the direction of current flow was reversed. The two abovementioned devices worked under AC; however, in both of these approaches, only half of the device surface emits light at the same time because they are equivalent to two LEDs connected in an antiparallel configuration.

In this article, we propose an alternative way to obtain an AC-driven device with embedded TJs, which allows us to use the full surface of the device for lighting. We will refer to this device as a bidirectional light emitting diode (BD LED) because the light is emitted from the same active region in both the positive and negative bias regime. BD LED consists of a single III-N QW surrounded by two oppositely oriented TJs. This design is advantageous, as it combines the simplicity of the symmetrical structure with the high quantum efficiency characteristic for devices based on III-nitrides. Furthermore, several BD LEDs can be stacked vertically to multiply the optical power

obtained from a single device. The optical and electrical properties of such BD LEDs under DC and AC are investigated.

It should be emphasized here that during the last decade III-N TJs have reached maturity and a high tunneling current under low operating voltage can be achieved[8,16–18]. TJ are used to stack several LEDs or laser diodes (LDs) for multiplying the optical power or obtaining multi wavelength optical spectrum[19–23], to control current flow in micro-LEDs[24–27], in distributed-feedback LDs[28] or in tunneling field-effect transistors[29]. Additionally, TJs can be used to invert the direction of current flow with respect to the arrangement of piezoelectric fields in the active region[30–33]. However, LEDs with only a single TJ (bottom or top) still resemble conventional LED structures, in which the QW is located inside the p-n junction of a LED. Such devices can only operate in one bias regime, i.e. the LED or LD is forward biased and TJ is reverse biased[8,17–28,30–33]. The unique design of our BD LED, in which a single QW is surrounded by two TJs, one from each side, allows for operation in both bias regime.

## Results
### BD LED structure

The structures of a single and a stack of two BD LEDs are presented in Fig. 1a, b, respectively. Samples were grown by plasma-assisted molecular beam epitaxy (PAMBE) on Ga-polar (0001) bulk GaN crystals. The structures are symmetrical with respect to the QW in regard to the arrangement of epitaxial layers and their doping levels. The growth was initiated with 200 nm thick GaN:Si (Si: $1 \times 10^{19}$ cm$^{-3}$) followed by heavily doped bottom n-p TJ, which consisted of 20 nm $In_{0.02}Ga_{0.98}N$:Si (Si: $1 \times 10^{20}$ cm$^{-3}$) and 10 nm $In_{0.02}Ga_{0.98}N$:Mg (Mg: $1 \times 10^{20}$ cm$^{-3}$). Then, a 35 nm thick p-type $In_{0.02}Ga_{0.98}N$:Mg region is placed, in which the Mg concentration decreases linearly from $1 \times 10^{20}$ cm$^{-3}$ to $2 \times 10^{19}$ cm$^{-3}$. In the middle of the BD LED structure, an unintentionally doped (UID) active region is located, which consists of a 10.4 nm wide $In_{0.17}Ga_{0.83}N$ quantum well (QW) surrounded by 30 nm $In_{0.08}Ga_{0.92}N$ barriers from each side. A relatively wide QW was chosen

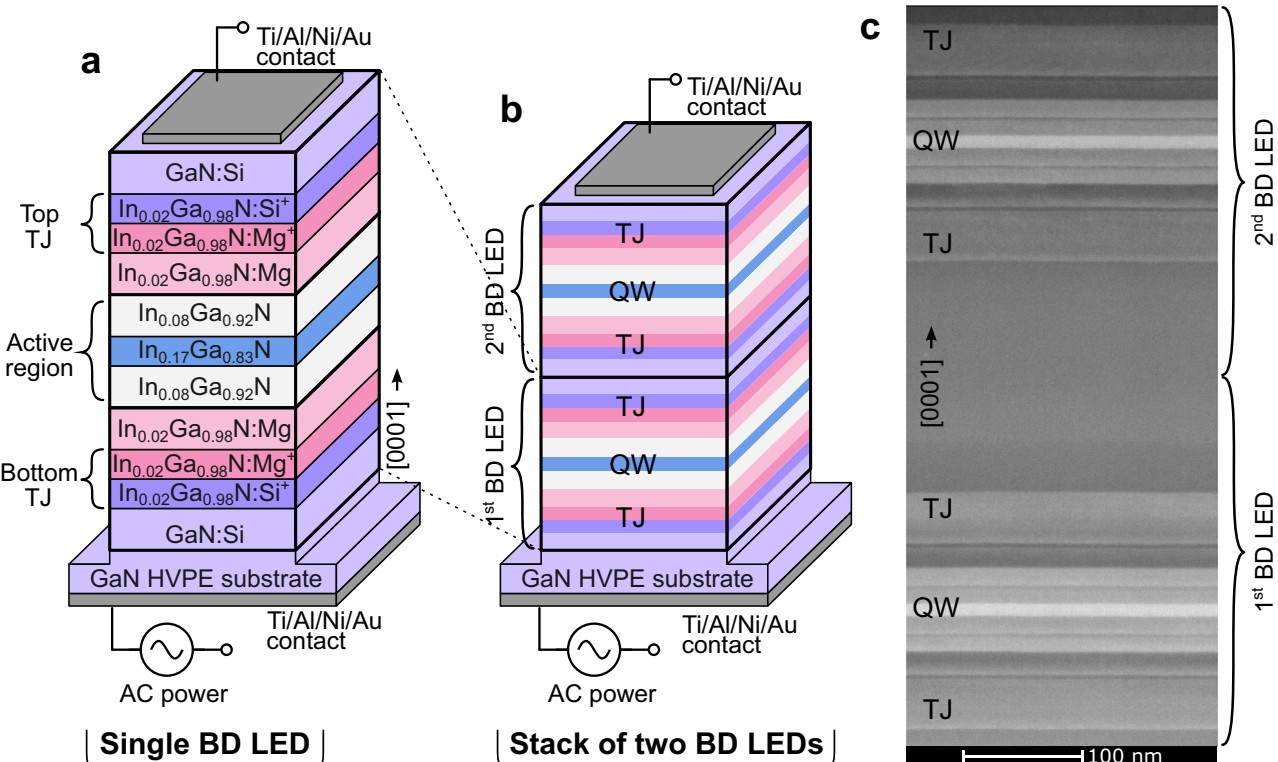

**Fig. 1 | The structure of bidirectional light emitting diode (BD LED). a** Schematic structure of single BD LED and **b** stack of two BD LEDs. **c** STEM cross-section image of the stack of two BD LEDs. Visible dark horizontal lines are growth interrupts.

due to a higher internal quantum efficiency than thin QWs[34,35]. Above the active region, symmetrically with respect to the QW, the same layer stack, but in the inverted order, is placed. The layers consist of a 35 nm $In_{0.02}Ga_{0.98}N$:Mg region with the Mg concentration increasing linearly from $2 \times 10^{19}$ cm$^{-3}$ to $1 \times 10^{20}$ cm$^{-3}$, followed by the top p-n TJ. The top TJ has the same chemical composition as the bottom TJ. The structure is then capped with 100 nm GaN:Si (Si: $3 \times 10^{19}$ cm$^{-3}$). In the case of the stack of two BD LEDs the sequence of layers repeats.

In the case of structures with buried p-type layers, PAMBE is more suitable because the p-type region is active immediately after growth. In contrast, for structures grown by metalorganic vapor phase epitaxy (MOVPE), the Mg-doped regions must be thermally annealed in order to remove hydrogen, which passivates p-type. However, vertical diffusion through the n-type region is suppressed[36,37]. Nevertheless, recently, two methods were proposed that allow for the activation of buried p-type layers grown by MOVPE but create some challenges for processing: activation by sidewalls and the selective area growth method[27,38].

The epitaxial structure of the stack of two BD LEDs is presented in the cross-section STEM image shown in Fig. 1c. The low number of extended defects and sharp interfaces shows that the device stacking process can be used for a larger number of BD LEDs. Such a good crystalline quality results from earlier investigations of highly doped TJs[18]. In that work, we found that doping levels up to $2 \times 10^{20}$ cm$^{-3}$ in TJ do not degrade the crystalline quality and give good electrical performance. We do not see any limitations on the number of BD LEDs stacked together, as the crystal quality of the structure remains very good[19,20]. Therefore, by stacking dozens of BD LEDs in one epitaxial process the resulting device can even be adapted to the supply voltage of the electrical grid.

We study the BD LED properties by analyzing its band diagrams. A simplified structure of unbiased single BD LED is presented in Fig. 2a, whereas Fig. 2b–d show its band diagram, electric field ($F$) together with polarization sheet charges ($\sigma$) and electron and hole concentrations ($n$, $p$), respectively. The active region of BD LED is located between two p-type regions, rather than between a p-type and a n-type, like in conventional LEDs. Although the epitaxial structure of the BD LED is symmetrical with respect to the QW, its band structure presented in Fig. 2b is not. This is a consequence of polarization charges (both spontaneous and piezoelectric) characteristic to growth performed on Ga-polar side (0001) GaN substrate[39]. Polarization sheet charges ($\sigma$), which appear at the interfaces where the In composition is changed are presented in Fig. 2c. The greatest influence of the polarization charges on to band arrangement is found in the active region. The Fermi-level on the left-hand side of the QW is pinned to the p-type by holes accumulated at $In_{0.08}Ga_{0.92}N/In_{0.17}Ga_{0.83}N$ interface with negative polarization charges (see Fig. 2c, d). In contrast, no carriers accumulate on the right-hand side of the QW at the $In_{0.17}Ga_{0.83}N/In_{0.08}Ga_{0.92}N$ interface. However, due to the positive polarization charges there is an electric field ($F$) of $-1.2$ MVcm$^{-1}$ present in the QW.

The TJs dedicated for BD LEDs are designed to have the same tunneling characteristics for the top and bottom configurations and to be transparent to blue light. Therefore, we use $In_{0.02}Ga_{0.98}N$ homojunctions in which the heavily doped region is much larger than the depletion width of the junction. Hence, the electric field ($F$) in both TJs originates only from the ionized dopants, as it is presented in Fig. 2c. Polarization sheet charges, that occur at GaN/$In_{0.02}Ga_{0.98}N$ and $In_{0.02}Ga_{0.98}N/In_{0.08}Ga_{0.92}N$ interfaces are located outside of depletion width of the heavily doped TJs regions and therefore they do not affect the tunneling for both bottom and top TJs. As a result the tunneling properties of both TJ are the same for both tunneling directions as shown in Supplementary Materials in Fig. S1.

In order to design transparent InGaN TJ, we consider two separate mechanisms that may introduce optical losses. The first is the

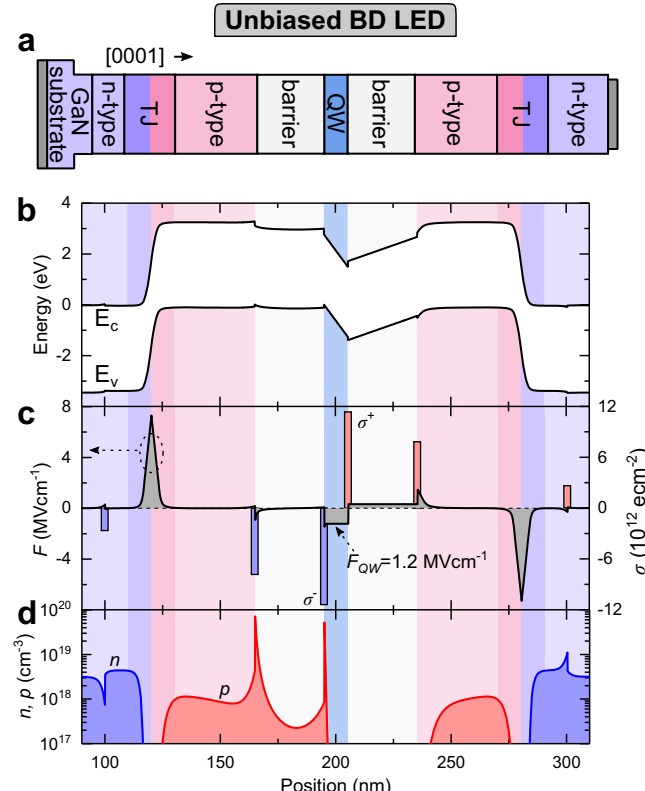

**Fig. 2 | Band diagram of unbiased BD LED. a** Schematic structure of a single BD LED. The corresponding **b** band diagram, **c** electric field ($F$) and polarization sheet charges ($\sigma$), **d** electron and hole carrier concentrations ($n$, $p$) present in the BD LED under zero bias.

band-to-band light absorption and the second is related to light absorption on dopants. In the case of BD LED presented in this paper band-to-band absorption on TJ is not present because we used $In_{0.02}Ga_{0.98}N$ in TJ and its bandgap energy is higher than energy of light emitted from $In_{0.17}Ga_{0.83}N$ QW. On the other hand, doping of the TJ at the level of $1 \times 10^{20}$ cm$^{-1}$ should not result in absorption coefficient higher than $1 \times 10^{3}$ cm$^{-1}$ [40,41]. The highly doped layers are extremely thin, which strongly limits absorption. The calculated transmittance of light travelling perpendicular to TJ is 99.4%. In case of conventional LEDs, the commonly used indium tin oxide (ITO) transparent contacts have an absorption coefficient higher than $1 \times 10^{3}$ cm$^{-1}$ [42]. Additionally, thickness of ITO is usually higher than that of TJ and the transmittance in highest efficiency LED is 95%[43]. Therefore, the use of TJ is not the limiting factor in the output power of BD LEDs.

## BD LED operating principle

The biasing convention of the BD LEDs is as follows: the BD LED is positively biased, when we apply a positive potential to the top electrical contact, and negatively, when we apply a negative potential, like it is schematically shown in Fig. 3a, d, respectively. Fig. 3b, e show the band diagrams of single BD LED under positive and under negative biases, respectively. Fig. 3c, f show the magnification of the band diagrams in the QW region. The band diagrams were calculated for current density of 1 kAcm$^{-2}$ and $-1$ kAcm$^{-2}$. The bottom TJ is located on the left-hand side, while the top TJ on the right-hand side of the active region. Both TJs are able to operate in two modes: carrier tunneling when reverse biased, or carrier drift-diffusion transport when forward biased. The biasing conditions of top or bottom TJ is imposed by the bias of the whole BD LED. Thanks to this property, the carriers are able to reach the same QW in both positive and negative biases of a single BD LED.

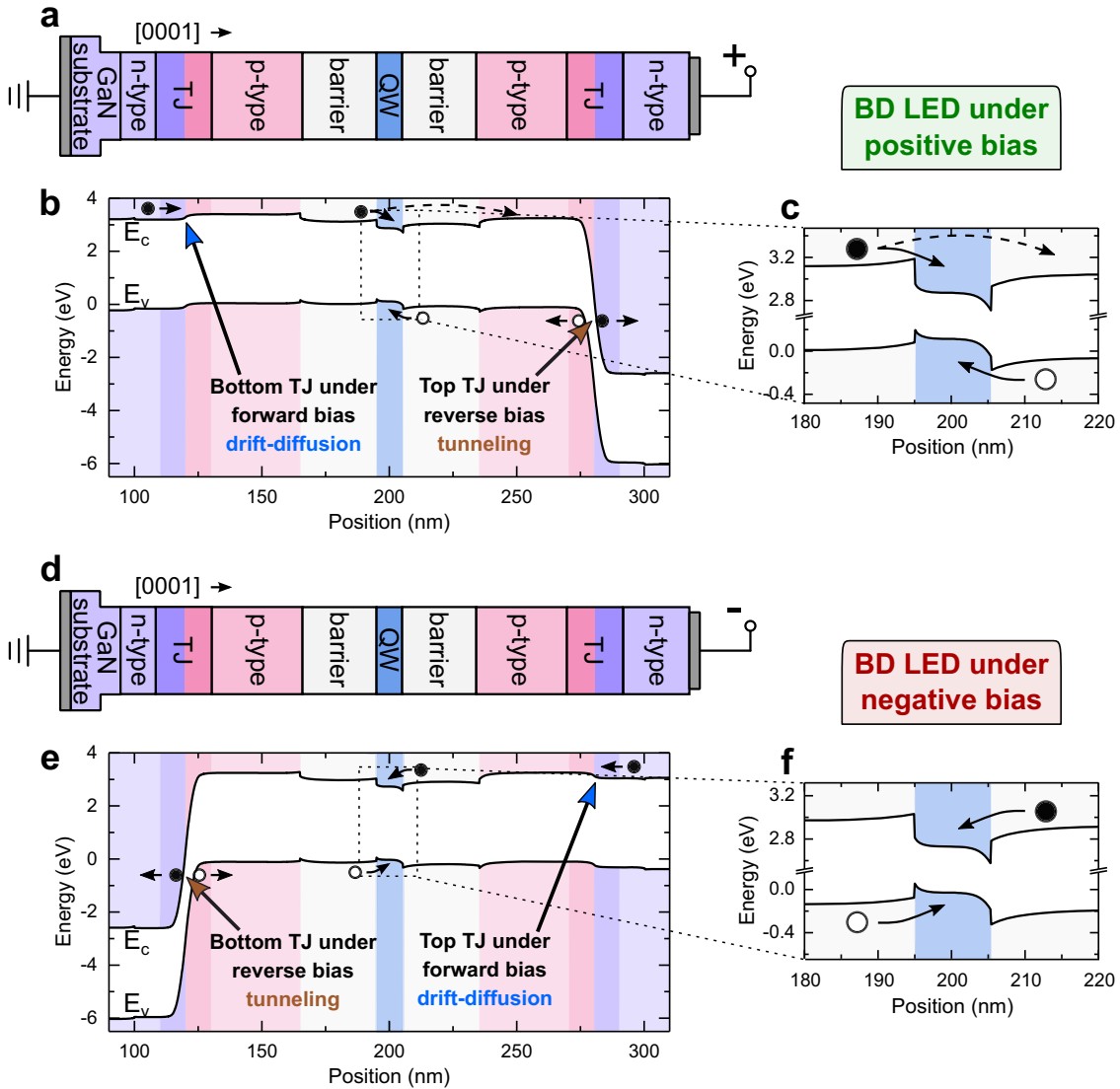

**Fig. 3 | Operating principle of BD LED under positive and negative bias. a, d** Power supply schemes, **b, e** band diagrams and **c, f** magnification of the active region of the BD LED under positive and negative biases, respectively. The band diagrams were calculated for a current densities of 1 kAcm$^{-2}$ and −1 kAcm$^{-2}$.

Under the positive bias of the BD LED (Fig. 3b), the top TJ is reverse biased, interband tunneling of carriers occurs, and thus, holes are injected into the QW from the right-hand side. Simultaneously, the bottom TJ is biased in the forward direction, so electrons are injected into the QW from the left-hand side. Contrary, for the negative bias of the BD LED (Fig. 3e), the bottom TJ is reverse biased, while the top TJ is biased in the forward direction. Therefore, the holes are injected from the left-hand side, and the electrons are injected from the right-hand side. In other words, when the BD LED is negatively biased, the sides from which the carriers are injected into the QW are interchanged. Importantly, in both bias cases (positive and negative), the carriers are injected into the same active region. Therefore, in our concept, the BD LED emits light from the entire surface of the device under AC conditions. This distinguishes the proposed device from the devices published in the literature in which only part of the device can emit light at the same time[11,12,14,15].

Utilization of two TJs, one on each side of the active region, makes the BD LED structure symmetrical. However, the growth on Ga-polar (0001) GaN substrate introduces an asymmetry in the QW, due to the arrangement of the built-in spontaneous and piezoelectric sheet charges[39]. The barrier for electron escape from the QW towards the right-hand side is lower than barrier for escape towards the left-hand

side as can be seen in Fig. 3c, f. When the positive bias is applied and the electrons are injected into the QW from the left-hand side, the low barrier can lead to electron escape from the QW to the right-hand side. Therefore, taking into account the direction of the carrier injection, the band alignment in the active region is more favorable for negative bias (see Fig. 3d–f) and leads to high injection efficiency. In the case of positively biased BD LED, the barrier for electron escape from the QW is small (Fig. 3c) and the resulting injection efficiency is low. The electrons, which overshoot the QW under positive bias, can give rise to parasitic recombination with holes outside of the QW. We discuss the effect of built-in electric field on the positively and negatively biased active region of the BD LED in more details in the Supplementary Materials in Fig. S2.

Furthermore, several BD LED can be stacked together in one epitaxial process in order to obtain higher optical power or multi-color emission. The principle of operation of stack of two or more BD LEDs is discussed in the Supplementary Materials in Fig. S3.

## BD LED properties under DC
The current-voltage (IV) characteristics of the examined devices are presented in Fig. 4a. For both positive and negative biases, the IV characteristics resemble the IV curves for the forward biased

conventional LED, and more importantly, the light is emitted above the turn-on voltages in both biasing conditions. The turn-on voltages are approximately 5 V and −5.5 V, while the operating voltage, for current densities of 1 kAcm⁻² and −1 kAcm⁻² are 6.9 V and −8 V, respectively.

The turn-on voltage of a BD LED is about 2 V higher than that of a conventional blue III-N LEDs. This is caused mainly by the additional voltage drop across the reverse biased TJ that operates in the tunneling mode. We calculated the tunneling current for both top and bottom TJs (presented in the Supplementary Materials in Fig. S1) with our recently developed tunneling model[18]. We estimate that the turn-on voltages on BD LED increases by 1.2 V due to voltage drop across the TJ that is polarized reverse. Moreover, calculations show that both TJs have identical tunneling IV characteristics. The remaining difference in

voltage may be due to the forward bias TJ. The turn-on voltage of a forward biased p-n junction depends on the energy gap of the semiconductor from which the junction is made. Therefore, a forward biased p-n junction made of $In_{0.02}Ga_{0.98}N$ will require a higher voltage than a conventional LED with an $In_{0.17}Ga_{0.83}N$ QW. Additional energetic barriers formed at the interfaces between the p-type region and the undoped $In_{0.08}Ga_{0.92}N$ layer may be source of asymmetry in the IV characteristic of BD LED.

In case of the stack of two BD LEDs, the turn-on voltages are 9.4 V and −11.2 V, for positive and negative biases, respectively. The operating voltage, for current densities of 1 kAcm⁻² and −1 kAcm⁻², are 12.5 V and −13.3 V, respectively. Stack of two BD LEDs operates at a slightly lower voltage than expected, when multiplying the voltage of the

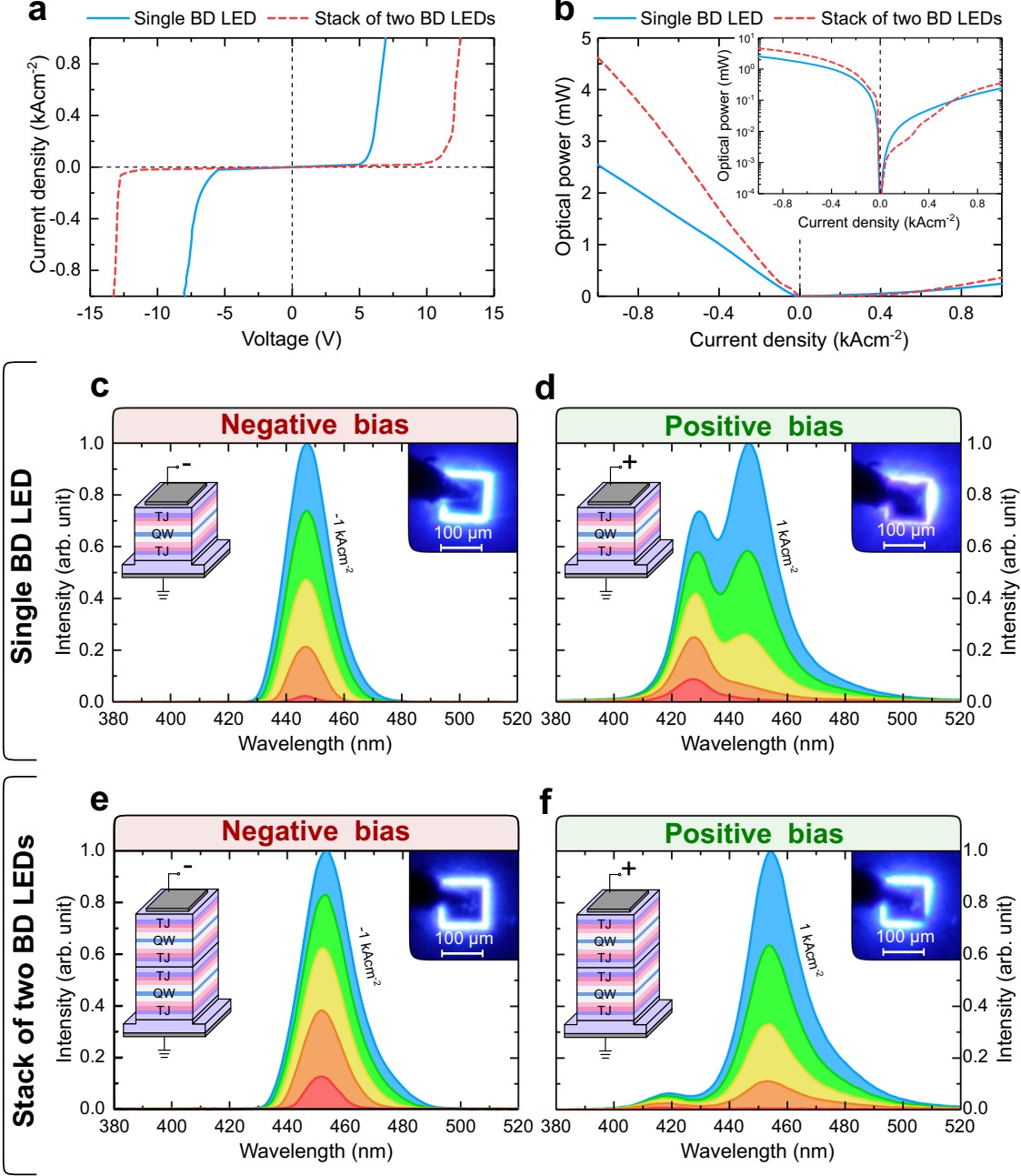

**Fig. 4 | Electrical and optical properties of single and stack of two BD LEDs.** **a** Current-voltage and **b** optical power characteristics of single and stack of two BD LEDs. Optical spectra of **c, d** single and **e, f** stack of two BD LEDs under negative and positive bias measured in steps of 0.2 kAcm⁻² up to ±1 kAcm⁻², respectively. Insets show real color pictures showing emission at a current density of ±1 kAcm⁻² obtained from devices with a size of 100 × 100 μm².

single BD LED by a factor of two. Indeed, the voltage on stack of two BD LEDs should be slightly lower than twice the value of voltage on a single device, because the resistances of substrate and metal contacts are common in both cases. Only the resistance of the diodes and TJs is doubled.

The dependence of optical power on current density is presented in Fig. 4b. Single BD LED reaches 2.5 mW and 0.24 mW of optical power, whereas stack of two BD LEDs reaches 4.6 mW and 0.36 mW at −1 kAcm$^{-2}$ and 1 kAcm$^{-2}$, respectively. The external quantum efficiency (EQE) is 0.9% and 0.09% at −1 kAcm$^{-2}$ and 1 kAcm$^{-2}$ for single BD LED. As already mentioned, the injection efficiency is higher for BD LED under negative than positive bias. We consider this to be the main reason of 10-times higher optical power of BD LED under the negative bias.

We would like to note that this BD LED has not yet been optimized for light extraction, so the results cannot be compared one-to-one with commercial LEDs. With processing that we used in this work, a lot of light is absorbed in thick metal contacts which cover the majority of the surface. Additionally, there is no encapsulations of the devices, which would enhance the extraction of light. We expect that after optimizing the light extraction from BD LEDs, we can achieve several times higher optical power. These steps are required to make BD LED comparable to well-developed conventional LEDs. In the case of stack of two BD LEDs the luminous power is roughly 1.8 times higher than in the case of single BD LED. This result shows that by stacking BD LEDs, one can increase the optical power obtained with the same supply current. The optical spectra of the examined devices were measured in steps of 0.2 kAcm$^{-2}$ up to ±1 kAcm$^{-2}$ and are shown in Fig. 4c–f. In the case of the single BD LED for negative bias, we observe only a single peak at λ = 447 nm related to recombination in the QW (Fig. 4c). However, for the positively biased BD LED, a second peak appears at λ = 428 nm. We attribute this parasitic peak to the recombination of electrons that overflow the active region, with holes in the In$_{0.02}$Ga$_{0.98}$N:Mg layer (in between 235 nm and 270 nm in the band diagram shown in Fig. 3b)[44,45]. This parasitic recombination dominates the spectra at low current densities, whereas at higher currents the QW peak prevails. In the case of the stack of two BD LEDs, the parasitic peak also occurs under a positive bias (at λ = 419 nm), but its contribution to the spectrum is much smaller. Importantly, under the negative bias, a single peak originating from the QW is present (at λ = 454 nm), similar to the single BD LED. The wavelengths of the peaks originating from the QWs in the case of single and stack of two BD LEDs are slightly different due to a small variation between the actual and intended indium contents in the QW during epitaxial growth. The insets of Fig. 4c–f present real color pictures of devices taken under a current density of ±1 kAcm$^{-2}$. In all cases, the light is emitted from the entire surface of the BD LED, but it is slightly more uniform for a negative bias. The dark square in the middle is the top metallization layer, which blocks part of the outgoing light.

In summary, we have noticed three main differences in operation between the positive and negative biases: (I) the operating voltage is higher at negative bias, (II) the optical power is 10-times lower at positive bias and (III) there is parasitic recombination at λ = 428 nm (at λ = 419 nm for the stack of two BD LEDs) at positive bias only. We attribute all of these differences to the presence of built-in electric fields in the BD LED structure grown on Ga-polar (0001) GaN substrate, which break its symmetry.

## BD LED properties under AC
The BD LED was tested under sinusoidal voltage conditions to check its capability as a direct AC-driven source of light. The single BD LED was powered with 12.6 V peak-to-peak alternating voltage with a frequency of 50 Hz. Simultaneously, the current passing through the device and light intensity were registered. All data are presented in Fig. 5a. The BD LED stays turned on for more than half of the total pulse period and emits light in both half-cycles. Although the light intensity was not

continuous, we did not observe any flickering with the naked eye. Due to these unique properties under AC, different than in other well-known semiconductor devices, we propose a special electronic circuit symbol for BD LEDs, which is shown in the insets of Fig. 5a.

Additionally, we performed a time-resolved electroluminescence measurement to experimentally determine the device response time from positive to negative operation modes. The single BD LED device was biased alternately with symmetrical supply voltage of 6.1 V and −6.1 V that corresponded to positive +100 Acm$^{-2}$ and negative −60 Acm$^{-2}$ pulses, both 5 μs long. First, the optical spectra were measured in the middle of both pulses and are shown in Fig. 5b. The measured spectra resemble those observed under DC conditions. For a current density of −60 Acm$^{-2}$, a single peak at λ = 447 nm is observed, whereas for 100 Acm$^{-2}$, two peaks appear at λ = 428 nm and λ = 447 nm. The time evolution of the light intensity together with the measured current density is presented in Fig. 5c. In Fig. 5d–e, we present in detail the region in which the direction of current flow changes. As can be observed, the current pulses are not ideally rectangular in shape. Indeed, current spikes are present in both cases and reach ±500 Acm$^{-2}$ in peak.

First, we will analyze the intensity of electroluminescence coming from carrier recombination in the QW at λ = 447 nm. When the current switches from 100 Acm$^{-2}$ to −60 Acm$^{-2}$ (Fig. 5d), the intensity of the QW peak immediately increases. Otherwise, when the current switches back again to 100 Acm$^{-2}$ (Fig. 5e), the intensity of the λ = 447 nm line decreases. However, the electroluminescence from the QW never goes away, and the BD LED switches very quickly. The response times from positive to negative pulses, and vice versa, are 60 ns and 30 ns, respectively, and they are comparable with the time needed to stabilize the current equal to 50 ns. Therefore, it seems that the presence of p-type layers surrounding the active region from both sides does not disturb the carrier transport. We will analyze this subject theoretically in the Discussion.

By examining the luminescence intensity of parasitic emission at λ = 428 nm, it can be seen that when the current goes to negative values, its intensity drops (see Fig. 5d), while when the current switches to positive values, the light intensity peaks (see Fig. 5e). Although, after stabilizing the current, the intensity of electroluminescence at λ = 428 nm is greater for −60 Acm$^{-2}$ than for 100 Acm$^{-2}$, one can notice that during the negative pulse, it does not form a distinct peak in the spectrum. This is just a tail of the QW peak, so in Fig. 5c–e, we have left the dashed line for this range.

## Influence of the QW polarization field on carrier injection
When the BD LED structure is biased, then it is interesting to consider the direction of carrier injection into the QW with respect to the arrangement of built-in electric fields. In these terms, for positive bias, the BD LED resembles conventional LEDs grown on Ga-polar substrate, while for negative bias it resembles LEDs grown on N-polar substrate. In such LEDs, the differences in the injection efficiencies for structures with various polarities have been widely discussed[30–33,46–48].

In the case of Ga-polar structures, low injection efficiency and electron overflow above the QW is a commonly known issue. This occurs due to the large difference in electron and hole mobility. This problem is obviated by the introduction of an electron blocking layer (EBL) on the p-side right after the QW[1]. A heavily doped AlGaN:Mg layer is most commonly used as the EBL, which forms an energetic barrier for electrons, due to its wide bandgap. When the EBL is not present or is not working properly (e.g. under cryogenic temperatures), electrons that pass over the QW recombine in the p-type region. This leads to parasitic recombination in the spectral range of λ = 420–430 nm[33,48]. In the literature, this peak is often identified as a transition from the unknown deep donor down to the Mg acceptor level[44,45].

On the other hand, in case of N-polar LEDs there is a favorable arrangements of built-in electric fields that stops electron overflow

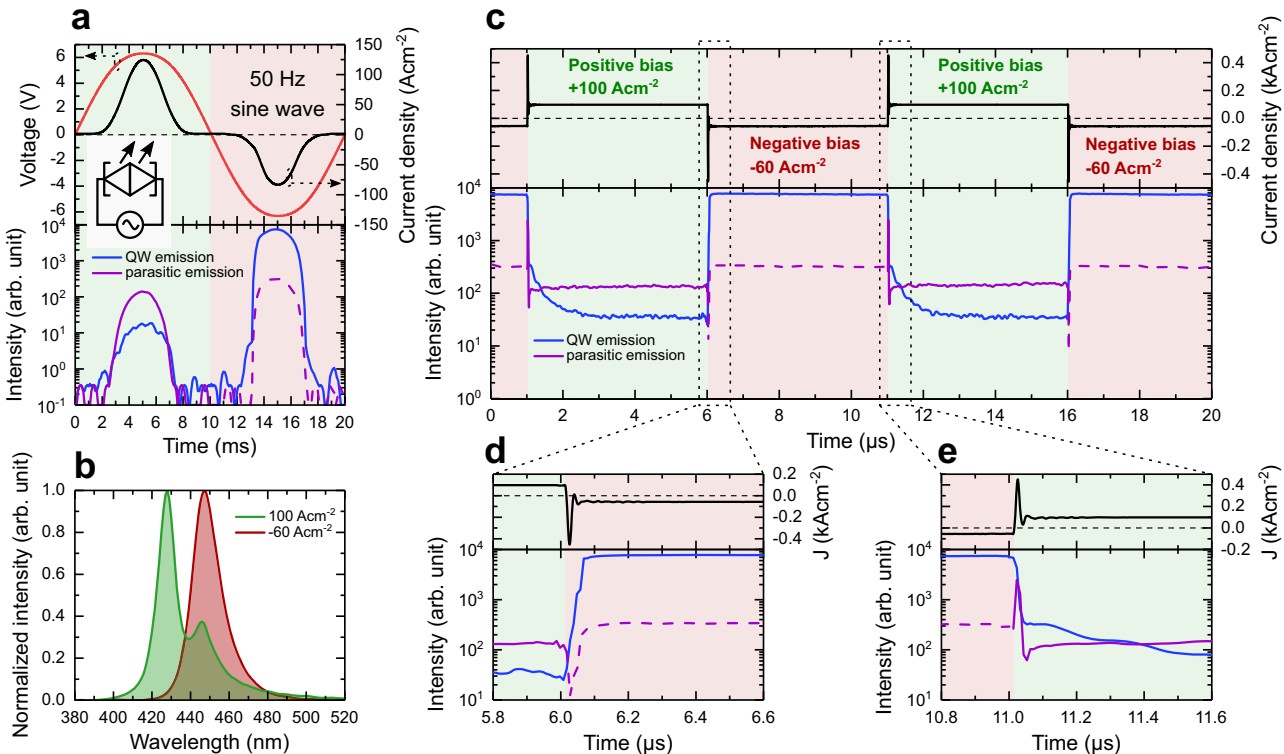

**Fig. 5 | Properties of BD LED under AC conditions. a** The current-voltage characteristics and electroluminescence intensity at λ = 447 nm (QW emission) and at λ = 428 nm (parasitic emission) of a single BD LED powered at 50 Hz and 12.6 V peak-to-peak alternating voltage. Insets show the proposed electrical circuit symbol of the BD LED connected to the AC source. **b** The single BD LED was examined under AC rectangular pulses of 100 Acm⁻² and −60 Acm⁻² each 5 μs long. The normalized optical spectra obtained in the middle of positive and negative pulses. **c** Time dependence of current density and electroluminescence intensity at λ = 428 nm and λ = 447 nm and **d, e** their magnification when current switches from positive to negative bias and vice versa, respectively.

over the QW[46,47]. Similarly, for negatively biased BD LED, the direction of carrier flow is reversed and electrons face a high energy barrier preventing them from leaving the QW. Indeed, in both cases, high carrier injection efficiencies are observed[31–33,46,47].

In view of these well-known properties of III-N LEDs, the electro-optical properties of BD LEDs under positive and negative biasing conditions correspond to Ga-polar and N-polar III-N LEDs, respectively. The injection efficiency is much higher for negatively biased BD LEDs, which leads to 10-times higher optical power in this case. Moreover, for the positively biased BD LED, electrons overflow above the QW and recombine with holes in the $In_{0.02}Ga_{0.98}N$:Mg layer (between 235 nm and 270 nm in the band diagram shown in Fig. 3b), as evidenced by the registered parasitic peaks at λ = 428 nm and λ = 419 nm for the single and stack of two BD LEDs, respectively. The electron overflow problem has been even more pronounced for pulse measurements presented in Fig. 5c–e, where the intensity of parasitic luminescence follows current spikes, especially when the BD LED was switched from negative to positive bias conditions.

Considering the BD LEDs presented here as direct AC-driven light sources, efforts should be made to increase the optical power obtained under positive bias. It should be pointed out that the asymmetry in the operation of the BD LEDs presented here is solely due to the existence of built-in electric fields specific to III-N structures grown on polar substrates. Fabricating a BD LED on a non-polar GaN substrate or in a different material system, in which the issue of embedded polarization charges does not exist, should give symmetric light-current-voltage (LIV) characteristics, regardless of the current flow direction. However, non-polar GaN substrates are much more expensive, and the growth is more challenging. Therefore, it is important to indicate the direction of research that would enhance the optical power obtained for positively biased BD LEDs grown on conventional Ga-polar (0001) substrates. It

would be beneficial to develop an EBL that, first of all, blocks the overflow of electrons for a positively biased BD LED, but also does not disturb the injection of electrons into the active region under a negative bias. Therefore, the AlGaN:Mg layer, which is typically used in conventional III-N LEDs as an EBL, might not be the best solution since it will form an energy barrier for electron injection for the negatively biased BD LED. Implementation of a suitable EBL layer to balance optical power will be the subject of future research on BD LEDs.

Once symmetrical light emission is obtained, when powered directly by AC, one will obtain twice as much light from a single BD LED than from a single conventional LED. This will be a step towards greater energy efficiency. It is worth highlighting the ability to stack multiple BD LEDs during a single epitaxial process to multiply the optical power obtained at the same current density. By growing a cascade of BD LEDs, high light output can be provided when operating at low peak current and high voltage, such as in an electrical grid[21].

### Electron transport through p-type layers in BD LEDs

For conventional LEDs, electrons are injected into the active region directly from the n-type doped region. However, in the case of BD LED, electrons first have to pass through a thin p-type region placed below and above the active region for positive and negative biases, respectively. Electrons that flow though the p-type region on the path to the QW may cause the ionization of neutral magnesium acceptors ($R_A^-$) following the formula $e^- + N_A^0 \rightarrow N_A^-$, which is shown in Fig. 6. Additionally, there are two other processes that are always present in the p-type region and occur between the acceptor level ($N_A$) and the valence band ($E_V$) (see Fig. 6). Neutral magnesium acceptors are thermally ionized, which promotes holes to the valence band ($R_T$), which is $N_A^0 \rightarrow N_A^- + h^+$. In the opposite process, ionized acceptors are neutralized by holes ($R_A^0$). $R_T$ and $R_A^0$ are in thermal equilibrium under

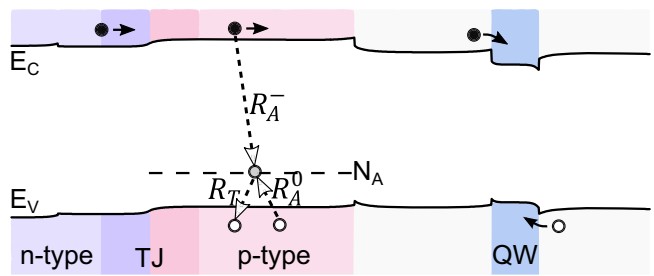

**Fig. 6 | Schematic representation of processes taking place in the left-hand side p-type region on the path of electrons injected into QWs for a positively biased BD LED.** $R_A^-$ is the ionization of the neutral magnesium acceptor by electrons, $R_T$ is the thermal ionization of the neutral acceptor that promotes holes to the valence band, and $R_A^0$ is the neutralization of the ionized magnesium acceptor by holes. The same processes occur in the right-hand side p-type region when the BD LED is negatively biased.

constant bias, providing a locally constant concentration of holes over time. However, if the rate of the $R_A^-$ process is much larger than $R_T$ and $R_A^0$, then all of the incoming electrons will first ionize neutral acceptors and only then will they reach the QW. Hence, it may be a source of unwanted loss of injected electrons and cause a delay in switching the device from negative to positive bias and vice versa.

The rates of the $R_A^-$, $R_T$, and $R_A^0$ processes can be determined from the following formulas[44]:

$$R_A^- = C_{nA}N_A^0 n \tag{1}$$

$$R_T = N_A^0 C_{pA} N_v g^{-1} \exp\left(-E_A/kT\right) \tag{2}$$

$$R_A^0 = C_{pA} N_A^- p \tag{3}$$

where $C_{nA}$ is the electron capture coefficient ($3.2 \times 10^{-12}$ cm³s⁻¹)[44], $N_A^0$ is the concentration of neutral magnesium acceptors, $n$ is the concentration of electrons in the conduction band, $C_{pA}$ is the electron capture coefficient ($5 \times 10^{-7}$ cm³s⁻¹)[44], $N_v$ is the effective density of states in the valence band, $g$ is the degeneracy factor of acceptor level, $E_A$ is the ionization energy of magnesium acceptor, $k$ is the Boltzmann constant, $T$ is temperature, $N_A^-$ is the concentration of ionized magnesium acceptors and $p$ is the concentration of holes in the valence band.

In general, considering all three processes, the kinetic equation describing the in-time changes in the concentration of neutral acceptors can be written as:

$$\frac{dN_A^0}{dt} = R_A^0 - R_T - R_A^- \tag{4}$$

However, while there is no electron in the p-type Eq. (4) reduces to:

$$\frac{dN_A^0}{dt} = R_A^0 - R_T \tag{5}$$

We will now consider the case, when a positive bias is suddenly applied to unbiased or negatively biased BD LED, following the experiment, to check if there is any significant delay in electroluminescence caused by $R_A^-$ process. In other words, the in-time stable Eq. (5) changes to Eq. (4). Now, we need to check if $N_A^0$ is still governed by thermodynamic processes $R_T$ and $R_A^0$. As an example, we will estimate their rates locally for a point in the middle of the considered p-type region at 142.5 nm. We extract the carrier concentrations in the structure using the 1-dimensional drift-diffusion charge control solver (1D-DDCC) made by Y-R Wu[49–51] and use constants in Eqs. (1–3) from the paper of Reshchikov et al.[44].

First, we calculate the rates of $R_T$ and $R_A^0$ for the unbiased structure, in which no electron current flows through the p-type region. Under these conditions, $p$, $N_A^0$ and $N_A^-$ were locally $1.0 \times 10^{18}$ cm⁻³, $6.5 \times 10^{19}$ cm⁻³ and $1.0 \times 10^{18}$ cm⁻³, respectively, which gives rates of $R_T$ and $R_A^0$ equal to $2.8 \times 10^{29}$ cm⁻³s⁻¹ and $5.3 \times 10^{29}$ cm⁻³s⁻¹, respectively. The order of magnitude of both processes is the same, whereas the small inequality of $R_A^-$ and $R_T$ values might result from the different carrier ionization energies assumed in the simulations (0.178 eV) and in the paper of Reshchikov et al. (0.15 eV)[44], which affects the calculated concentration of holes. Then, by applying the positive bias of 100 Acm⁻², we obtain $n$ of $1.94 \times 10^{14}$ cm⁻³, which gives the rate of $R_A^-$ equal to $4.06 \times 10^{22}$ cm⁻³s⁻¹.

The above calculations show that the rate of $R_A^-$ is seven orders of magnitude smaller than $R_A^0$ and $R_T$. This means that the $R_A^-$ process is too weak to effectively ionize the neutral magnesium acceptors. $N_A^0$ does not change even if electrons are passing through p-type. Therefore, we do not observe any significant time delay due to the electron-acceptor ionization process $R_A^-$ from switching the current direction of the BD LED in the emission from the QW. Otherwise, if the $R_A^-$ process were dominant, we should observe delays of hundreds of nanoseconds, which is the time needed to supply a matching number of electrons equal to the total number of neutral magnesium acceptors in the p-type region between 120 nm and 165 nm.

Nevertheless, the relatively small value of the $R_A^-$ process does not mean that it can be completely ignored in regard to the loss of electron current ($j_{\mathrm{loss}}$) passing through Mg-doped layers. Current losses can be estimated from the equation:

$$j_{\mathrm{loss}} = q \int_{x_1}^{x_2} R_A^-(x)dx \tag{6}$$

where $q$ is the elementary charge, $x_1$ and $x_2$ are the start and end points of the considered p-type region (120 nm and 165 nm) and $R_A^-(x)$ is the rate of the space-dependent electron-acceptor ionization process. From Eq. (6), we obtain $j_{loss}$ equal to 0.03 Acm⁻², which is, however, negligibly small compared to the total current of 100 Acm⁻². From the above discussion, it is clear that the presence of p-type layers surrounding the active region has no significant impact on electron transport to the QW, which has also been confirmed by the time-resolved electroluminescence experiment. This makes BD LEDs feasible even in applications requiring fast switching times on the order of tens of nanoseconds.

## Discussion

This paper provides insights into direct AC-driven semiconductor light sources. Here, we demonstrated a proof-of-concept device – the GaN-based bidirectional light-emitting diode (BD LED), which consists of a single III-N wide QW surrounded by two oppositely oriented TJs. This allows the injection of electrons and holes into the QW during both positive and negative biases. We have successfully shown that BD LEDs are able to operate and emit light in both directions of the supplied current, especially in both half-cycles of the AC power supply. No delay in electroluminescence was observed after switching the current direction, which indicates that thin p-type layers surrounding the active region do not affect the transport of electrons. The stack of two BD LEDs is demonstrated to show future possibilities to vertically interconnect many devices in one epitaxial process in order to multiply the light emitted from a single device and to withstand high-voltage application. Moreover, an interesting feature of such a system is color mixing to produce white light. This can be done by varying the In content of the QWs in subsequently grown BD LEDs in stack.

In addition, BD LEDs can be considered as a platform for investigating the influence of built-in electric fields on the electro-optical properties of III-N devices. Indeed, we observed different optical powers under positive and negative biases, even though recombination occurred from the same QW. This reflects different carrier

injection efficiencies for different current directions. Furthermore, to make the III-N BD LED grown on a Ga-polar substrate more attractive as a direct AC-driven light source, we recommend breaking the symmetry of the structure. We propose to increase the optical power for a positive bias by introducing a specially designed EBL on the right side of the active region. We believe that with the unique construction of BD LEDs, it will be possible to design sophisticated experiments to better understand the physical processes occurring in III-N LEDs in general.

## Methods

The structures were grown in metal-rich conditions using plasma-assisted molecular beam epitaxy (PAMBE) on the Ga-polar (0001) side of n-type conductive ($n \approx 10^{18}$ cm$^{-3}$) commercially available GaN bulk crystals obtained using hydride vapor phase epitaxy. The growth was performed in a VG Semicon V90 reactor. GaN layers were grown at 730 °C with a growth rate of 0.36 μmh$^{-1}$. InGaN layers were grown at 650 °C with growth rates of 0.76 μmh$^{-1}$ and 0.85 μmh$^{-1}$ for QW and other InGaN layers, respectively. The growth temperature was controlled in situ by laser reflectometry[52]. Growth interrupts, which are visible in STEM presented in Fig. 1c as dark horizontal lines, were performed to control metal coverage during InGaN growth. The indium content of the InGaN layers was controlled by adjusting the gallium to nitrogen ratio according to the model presented in the paper of Turski et al.[53]. The approximate growth duration of a single BD LED structure, including the time necessary to stabilize growth condition in the reactor, was 2 h. Therefore, the time needed to grow a stack of n BD LEDs will be n times 2 h. Details of PAMBE growth can be found elsewhere[54].

Samples were processed into devices with dimensions of 100 × 100 μm$^2$ by standard lift-off photolithography process using LaserWriter. First, a negative photoresist NLOF 2020 was spin coated onto the samples and rows of 100×100 μm$^2$ squares were exposed. The mesa was etched using chlorine inductively coupled plasma reactive ion etching (ICP RIE) to a depth of 450 nm and 900 nm for single and stack of two BD LEDs, respectively. The photoresist was removed in dimethyl sulfoxide (DMSO) and a second photolithography process was carried out in which 90 x 90 μm$^2$ windows were opened at the top of each mesa, while the rest of the etched samples and sides of the mesa were protected. Next, a Ti/Al/Ni/Au (30/60/40/75 nm) metallic contact was sputtered on the top of sample, the photoresist was removed in dimethyl sulfoxide (DMSO) and then the metallic contact was annealed at 750 °C for 1 min in an N$_2$ atmosphere. The bottom metallization was applied to the entire reverse N-side surface of the sample and was not annealed.

The electro-optical properties of BD LEDs under DC conditions were examined with a source meter unit (Keithley, SMU2400) and a spectrometer (Ocean Optics, USB4000). Dozens of devices were measured for both single and stack of two BD LEDs, and representative ones were selected for further analysis. In order to measure the optical power the devices were cleaved, mounted and measured in a calibrated integrating sphere (Thorlabs, S142C).

The AC time-resolved electroluminescence measurement was performed in homemade experimental setup, which combines arbitrary waveform generator (Siglent, SDG2042X), oscilloscope (Tektronix, TDS3032B) and spectrometer (SPEX Industries, 1000 M) coupled with photomultiplier tube (Hamamatsu, R636-10) and photon counter (Stanford Research Systems, SR400). The BD LED was powered by arbitrary waveform generator (Siglent SDG2042X), which was the source of sinusoidal and rectangular voltage pulses. Current, which flowed through the device, was in time controlled with the oscilloscope by checking the voltage drop on a 50 Ohm resistor connected in series with the BD LED. The spectrophotometer was operated in two modes: spectral measurement at a specific moment in time with 1 nm resolution or real-time light intensity measurement for a specific wavelength in 10 ns time windows. The obtained data were smoothed with the weighted adjacent-averaging method, and the delay time was corrected based on the photomultiplier tube response time and the optical path of the light beam.

## Data availability

The raw datasets generated during and/or analyzed during the current study are available from the corresponding author on request. The processed data are provided with this paper in the Source Data file. Source data are provided with this paper.

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

## Acknowledgements

This work was supported partially by TEAM-TECH POIR.04.04.00-00-210C/16-00 (C.S.) and HOMING POIR.04.04.00-00-5D5B/18-00 (H.T.) projects of the Foundation for Polish Science co-financed by the European Union under the European Regional Development Fund and the Polish National Centre for Research and Development Grants LIDER/29/0185/L-7/15/NCBR/2016 (H.T.) and LIDER/35/0127/L-9/17/NCBR/2018 (G.M.) and National Science Center Poland within grant No. 2019/35/D/ST3/03008 (G.M.).

## Author contributions

M.Ż. conceived the idea of BD LEDs presented in this work. M.Ż. and G.M. established the BD LED epitaxial structure and designed the experiments. M.S. performed the BD LED epitaxy process. A.B., M.Ż., and P.W. contributed to BD LED measurements under AC conditions and time-resolved electroluminescence experiment. M.Ż., M.H., M.C. contributed to BD LEDs measurements under DC conditions. K.N.-S. performed processing of BD LEDs. A.L. performed the STEM analysis of the BD LEDs. M.Ż. and M.C. visualized the data in this article. M.Ż. and G.M. analyzed data and prepared the manuscript. C.S. and H.T. revised the manuscript and provided suggestions. All authors discussed the results and commented on the manuscript.

## Competing interests

A patent application describing the BD LEDs, presented in this work, has been filed to the Polish Patent Office (application number P.438744) and to the European Patent Office (application number EP22461527.8). M.Ż., G.M., M.S., K.N.-S., H.T., C.S. are co-inventors. The remaining authors declare no competing interests.
