## [Peer Review File · Nature Communications]

REVIEWER COMMENTS

Reviewer #2 (Remarks to the Author):

The paper titled “Bidirectional LED as an AC driven visible-light source” by Mikolaj Zak et al. explored a novel approach to demonstrate alternating current (AC) driven light-emitting diodes grown on GaN single crystal wafers. The presented method is exciting and clever, and the work is comprehensive. However, I have some general comments for the improvement and clarity of the manuscript.

1. The authors explored both single and stacked tunnel junction (TJ) LEDs. It is unclear as it was not compared with the conventional LED (without tunnel junction), how TJ LED would perform compared to conventional LED. The tunnel junction absorbs a lot of light, and the light intensity can be lower than the conventional LED.
2. The tunnel junction works better in reverse bias; when the tunnel junction is reverse biased, the LED is forward or positively biased. Therefore, the top TJ is forward-biased (Fig. 2d), and there is no tunneling only carrier diffusion through the top TJ. From my point of view, the top TJ is useless, it will not add carrier as there is no tunneling, and would just add extra voltage drop. For the better understanding, Fig. 2d can be considered like as (TJ (resistor+LED (p-i-n)
3. I think the paper can explain the biasing convention and the polarity clearly.
4. I assume the structures are Ga-face polar. In that case, the spontaneous+piezoelectric polarization fields in the bottom and top tunnel junction would be opposite. Therefore, the electric field for tunneling would be opposite, and one of the tunnel junctions would be less functional or useful for carrier recycling/tunneling.
5. Can the authors add one or more polarization and electric field direction schematics or band diagrams?
6. I do not understand what is opening voltage, is it “turn on” voltage’?
7. Can the authors provide any absolute number of light output power instead of an arbitrary unit?
8. In lines 141-142, “in the case of the single BD LED, the collected light power grows approximately linearly with the positive and negative supply currents”. I do not understand this statement. Since the TJ works better in reverse bias, this is why the LED where the TJ is reverse biased works better (10 times higher luminous power). Overall, the paragraph is hard to follow (lines 141-148)
9. It is unclear why the BD LED stays turned on just more than half the total pulse period? I think the BD LED is much brighter in one half (when the TJ is reverse biased and current tunneling happens), whereas in the other half, it should be much dimmer as TJ is forward biased and no tunneling occurs. It should be much brighter if it works with one LED but will not be bidirectional.
10. The statement in lines 231-232, “In fact, the positive and negative bias conditions of the BD LED can be directly compared to III-N LEDs grown on Ga-polar and N-polar GaN substrates, respectively.” This

statement is not valid. It cannot be the same because the polarization directions and spontaneous and polarizations fields will differ for each layer. This statement should be clarified.

11. How many LEDs can be stacked by TJ using planar growth process? The overall voltage of the stacked device is still low (maybe there are some leakage currents).

12. Can the authors add more details about the growth of TJ as well as LED? It would be interesting to see the growth rate/duration for the stacked structures.

13. Why were asymmetric positive 100 A/cm² and negative -60 A/cm² applied for the time-resolved electroluminescence measurement?

14. Overall, the writing can be improved; the manuscript is hard to read.

Reviewer #3 (Remarks to the Author):

This manuscript studies a bidirectional vertical InGaN LED with two TJs which can be AC driven which is interesting. Though there are several concerns from this study:

1) The TEM image in Fig. 1(c) is unclear. The contrast from TJs and active region is not good enough. It will be better if an improved TEM can be provided.

2) Also, there has been studies on the use of bottom and top TJs - how would this work compare to those existing works? What are the novelties?

3) How are the results in Fig. 2 generated? Please describe. How are the TJs designed for this LED? For example, what are the roles of thickness, doping, etc.?

4) It seems like the turn on voltage is very large from Fig. 3. Please explain.

5) Please also explain the dual peaks from Figs. 3(d) and 3(f).

6) Please describe the LED fabrication process in more details.

7) How does the power/EQE compare to other blue LEDs? Please provide a comparison discussion.

8) Please provide a discussion and figure on the internal field of the QWs for under forward bias vs. reverse bias. It's helpful to expand the physics of the device under AC condition.

9) Other than using polarized substrate, are there any ways to use this structure on c-plane GaN substrate to achieve more symmetrical performances?

Response to reviews of manuscript “Bidirectional LED as an AC driven visible-light source”

We appreciate the insightful feedback provided by the reviewers on our manuscript titled “Bidirectional LED as an AC-driven visible-light source”. We have carefully addressed reviewers comments and suggestions in our revised manuscript. The reviewers input, particularly regarding a more comprehensive description of the bidirectional LEDs operational principles and a deeper exploration of their electro-optical properties, has greatly contributed to the manuscript's improvement.

Our revised manuscript now presents a clearer exposition of the physics underlying bidirectional LEDs, enhancing its accessibility to potential readers. To accommodate the additional data generated during our revision process, we have included supplementary files labeled *Supplementary Files BD LED*. All changes to the original manuscript are marked as track changes within the file *BD LED manuscript revised*. Below, we have diligently responded to each point raised by the reviewers in the file *BD LED reviewers remarks*.

Reviewer #2 (Remarks to the Author):

The paper titled “Bidirectional LED as an AC driven visible-light source” by Mikolaj Zak et al. explored a novel approach to demonstrate alternating current (AC) driven light-emitting diodes grown on GaN single crystal wafers. The presented method is exciting and clever, and the work is comprehensive. However, I have some general comments for the improvement and clarity of the manuscript.

1. The authors explored both single and stacked tunnel junction (TJ) LEDs. It is unclear as it was not compared with the conventional LED (without tunnel junction), how TJ LED would perform compared to conventional LED. The tunnel junction absorbs a lot of light, and the light intensity can be lower than the conventional LED.

Indeed, InGaN TJ could introduce optical losses based on two separate mechanisms. The first is the band-to-band absorption and the second is related to absorption on dopants. In the case of BD LED presented in this paper band-to-band absorption on TJ is not present because we used $\text{In}_{0.02}\text{Ga}_{0.98}\text{N}$ in TJ and its bandgap energy is higher than energy of light emitted from $\text{In}_{0.17}\text{Ga}_{0.83}\text{N}$ QW. On the other hand doping of the TJ at the level of $1 \times 10^{20} \text{ cm}^{-3}$ should not result in absorption coefficient higher than $1 \times 10^3 \text{ cm}^{-1}$ [1]. The highly doped layers are extremely thin, which strongly limits absorption. The calculated transmittance of light travelling perpendicular to TJ is 99.4%. In case of conventional LEDs, the commonly used indium tin oxide transparent contacts have an absorption coefficient higher than $1 \times 10^3 \text{ cm}^{-1}$ [2]. Additionally, thickness of ITO is usually higher than that of TJ and the transmittance in highest efficiency LED is 95% [3]. Therefore, the use of TJ is not the limiting factor in the output power of BD LEDs. It was already shown by E. Young et. al., that efficient and transparent TJ can be obtained in GaN-based LED [4]. They present that external quantum efficiency (EQE) of TJ-LED was higher than for conventional LED with ITO p-type contact. Consequently, more important than light absorption at the TJ seems to be the increase of light extraction efficiency from the diode structure, after which TJ-LEDs can reach above 70% of EQE, as shown by Yonkee et. al. [5].

Changes in manuscript:

(lines 144-162)

The TJs dedicated for BD LEDs are designed to have the same tunneling characteristics for the top and bottom configurations and to be transparent to blue light. Therefore, we use $\text{In}_{0.02}\text{Ga}_{0.98}\text{N}$ homojunctions in which the heavily doped region is much larger than the depletion width of the junction. Hence, the electric field (F) in both TJs originates only from the ionized dopants, as it is presented in Fig. 2c. Polarization sheet charges, that occur at $\text{GaN}/\text{In}_{0.02}\text{Ga}_{0.98}\text{N}$ and $\text{In}_{0.02}\text{Ga}_{0.98}\text{N}/\text{In}_{0.08}\text{Ga}_{0.92}\text{N}$ interfaces are located outside of depletion width of the heavily doped TJs regions and therefore they do not affect the tunneling for both bottom and top TJs. As a result the tunneling properties of both TJ are the same for both tunneling directions as shown in Supplementary Materials in Fig. S1.

In order to design transparent InGaN TJ, we consider two separate mechanisms that may introduce optical losses. The first is the band-to-band light absorption and the second is related to light absorption on dopants. In the case of BD LED presented in this paper band-to-band absorption on TJ is not present because we used $\text{In}_{0.02}\text{Ga}_{0.98}\text{N}$ in TJ and its bandgap energy is higher than energy of light emitted from $\text{In}_{0.17}\text{Ga}_{0.83}\text{N}$ QW. On the other hand, doping of the TJ at the level of $1 \times 10^{20} \text{ cm}^{-3}$ should not result in absorption coefficient higher than $1 \times 10^3 \text{ cm}^{-1}$ ^{40,41}. The highly doped layers are extremely thin, which strongly limits absorption. The calculated transmittance of light travelling perpendicular to TJ is 99.4%. In case of conventional LEDs, the commonly used indium tin oxide (ITO) transparent contacts have an absorption coefficient higher than $1 \times 10^3 \text{ cm}^{-1}$ ⁴². Additionally, thickness of ITO is usually higher than that of TJ and the transmittance in highest efficiency LED is 95% ⁴³. Therefore, the use of TJ is not the limiting factor in the output power of BD LEDs.

2. The tunnel junction works better in reverse bias; when the tunnel junction is reverse biased, the LED is forward or positively biased. Therefore, the top TJ is forward-biased (Fig. 2d), and there is no tunneling only carrier diffusion through the top TJ. From my point of view, the top TJ is useless, it will not add carrier as there is no tunneling, and would just add extra voltage drop. For the better understanding, Fig. 2d can be considered like as (TJ (resistor+LED (p-i-n)).

Reviewer is correct, when we operate BD LED only under negative bias, the top TJ, which is forward biased, seems to be unnecessary (see Fig. R1e). On the other hand, for positively biased BD LED the bottom TJ seems to be unnecessary (see Fig. R1b). However, without these TJs the device would not operate under AC in both biasing regimes. Thanks to the fact that both TJ are present the device conducts current and emits light for both positive and negative voltage half-cycles.

To obtain the bidirectional properties of LED we made use of dual behavior of TJs that are: carrier tunneling, when TJ is reverse biased and carrier drift-diffusion transport, when TJ is forward biased. Thanks to this property, the carriers are able to reach the same QW in both positive and negative bias of single BD LED. To emphasize this fact, we show the band structure of the BD LED under positive and negative bias on Fig. R1, which can be found in the revised manuscript as Fig. 3.

Fig. R1 (Fig. 3 in revised manuscript) (a, d) Power supply schemes, (b, e) band diagrams and (c, f) magnification of the active region of the BD LED under positive and negative biases, respectively. Arrows present the direction of electron and hole currents. The band diagrams were calculated for a current density of 1 and -1 kA/cm².

Changes in manuscript:

(lines 178-182)

The bottom TJ is located on the left-hand side, while the top TJ on the right-hand side of the active region. Both TJs are able to operate in two modes: carrier tunneling when reverse biased, or carrier drift-diffusion transport when forward biased. The biasing conditions of top or bottom TJ is imposed by the bias of the whole BD LED. Thanks to this property, the carriers are able to reach the same QW in both positive and negative biases of a single BD LED.

3. I think the paper can explain the biasing convention and the polarity clearly.

Indeed, the biasing convention and polarity might not have been clear enough. We added schematics and keep to “positive bias” and “negative bias” for the BD LED device. The BD LED is positively biased when we apply a positive potential to the top electrical contact and negatively when we apply a negative potential there, like it is schematically shown in Figs. R1a and R1d. Moreover, we use “forward”

and “reverse” while referring to the bias of each TJ (i.e. in the positively biased BD LED, the bottom TJ is forward biased and the top TJ is reverse biased).

We made changes in section *BD LEDs operating principle*, to clarify BD LED biasing convention. We have added the biasing scheme for the BD LED diagram for the BD LED structure to the image in Fig. 3 of revised manuscript (see Fig. R1). We unified the color coding to the Figs. 3, 4, 5 of revised manuscript (here shown as Figs. R1-R3), in which green indicates positive bias and red indicates negative bias. In addition, we added two band diagrams of stacked BD LEDs operating under negative and positive bias to supplementary materials in Fig. S3 (here shown as Fig. R4).

Fig. R2 (Fig. 4 in revised manuscript) (a) Current-voltage and (b) optical power characteristics of single and stack of two BD LEDs. Optical spectra of (c-d) single and (e-f) stack of two BD LEDs under negative and positive bias measured in steps of 0.2 kA/cm^2 up to 1 kA/cm^2 , respectively. Insets show real color pictures showing emission at a current density of $\pm 1 \text{ kA/cm}^2$ obtained from devices with a size of $100 \times 100 \text{ }\mu\text{m}^2$.

Fig. R3 (Fig. 5 in revised manuscript) (a) The current-voltage characteristics and electroluminescence intensity at $\lambda=447$ nm (QW emission) and at $\lambda=428$ nm (parasitic emission) of a single BD LED powered at 50 Hz and 12.6 V peak-to-peak alternating voltage. Insets show the proposed electrical circuit symbol of the BD LED connected to the AC source. The single BD LED was examined under AC rectangular pulses of 100 A/cm² and -60 A/cm² each 5 μ s long. (b) The normalized optical spectra obtained in the middle of positive and negative pulses. (c) Time dependence of current density and electroluminescence intensity at $\lambda=428$ nm and $\lambda=447$ nm and (d,e) their magnification when current switches from positive to negative bias and vice versa, respectively.

Fig. R4 (Fig. S3 in Supplementary Materials) (a, c) Power supply schemes and (b, d) band diagrams of stacks of two BD LEDs under positive and negative bias regime, respectively, calculated for current density of 1 kA/cm^2 and -1 kA/cm^2 . Arrows present the direction of electron and hole currents.

Changes in manuscript:

(lines 173-175)

The biasing convention of the BD LEDs is as follows: the BD LED is positively biased, when we apply a positive potential to the top electrical contact, and negatively, when we apply a negative potential, like it is schematically shown in Figs. 3a and 3d, respectively.

(lines 217-219)

Furthermore, several BD LED can be stacked together in one epitaxial process in order to obtain higher optical power or multi-color emission. The principle of operation of stack of two or more BD LEDs is discussed in the Supplementary Materials in Fig. S3.

Supplementary Materials:

Fig. S3 presents the band diagram of a stack of two BD LEDs fabricated in a single epitaxial process. The carrier flow inside each section from the stack is the same as in a single BD LED. However, in this case, the current flow between each individual BD LEDs in the stack is important. For both positive and negative bias it is only the electron current.

4. I assume the structures are Ga-face polar. In that case, the spontaneous+piezoelectric polarization fields in the bottom and top tunnel junction would be opposite. Therefore, the electric field for tunneling would be opposite, and one of the tunnel junctions would be less functional or useful for carrier recycling/tunneling.

It is true that structures were grown on the Ga-face polarity of crystal, so it is necessary to consider the effect of built-in electric fields (spontaneous and piezoelectric) on TJ properties. In the presented BD LEDs TJs are composed from $\text{In}_{0.02}\text{Ga}_{0.98}\text{N}$ heavily doped with Si (20 nm) and Mg (10 nm) both at the level of $1 \times 10^{20} \text{ cm}^{-3}$. This 30 nm heavily doped region is larger than the depletion width of TJ, so the electric field (F) at the junction is only the field originating from the ionized dopants, as it is presented on Fig. R5c.

Spontaneous and piezoelectric charges, that occur at $\text{GaN}/\text{In}_{0.02}\text{Ga}_{0.98}\text{N}$ and $\text{In}_{0.02}\text{Ga}_{0.98}\text{N}/\text{In}_{0.08}\text{Ga}_{0.92}\text{N}$ interfaces are located outside of the junction region (see Fig. R5c). Therefore, these charges do not influence the electric field of the TJ and have no impact on tunneling currents and tunneling efficiency for both bottom and top TJs. To validate our considerations, we calculated tunneling currents using a previously developed model that takes into account the spontaneous and piezoelectric charges [6]. Fig. R6, shows the tunneling current as a function of voltage for bottom and top TJ used in examined BD LEDs. As can be seen the calculated tunneling currents are identical for both TJs. We present this chart in the supplementary materials as Fig. S1.

Fig. R5 (Fig. 2 in revised manuscript) (a) Schematic structure of a single BD LED. The corresponding (b) band diagram, (c) electric field (F) and polarization sheet charges (σ), (d) electron and hole carrier concentration (n , p) present in the BD LED under zero bias.

Fig. R6 (Fig. S1 in supplementary materials) Calculated tunneling current for bottom and top TJs of BD LED.

Changes in manuscript:

(lines 144-151)

The TJs dedicated for BD LEDs are designed to have the same tunneling characteristics for the top and bottom configurations and to be transparent to blue light. Therefore, we use $\text{In}_{0.02}\text{Ga}_{0.98}\text{N}$ homojunctions in which the heavily doped region is much larger than the depletion width of the junction. Hence, the electric field (F) in both TJs originates only from the ionized dopants, as it is presented in Fig. 2c. Polarization sheet charges, that occur at $\text{GaN}/\text{In}_{0.02}\text{Ga}_{0.98}\text{N}$ and $\text{In}_{0.02}\text{Ga}_{0.98}\text{N}/\text{In}_{0.08}\text{Ga}_{0.92}\text{N}$ interfaces are

located outside of depletion width of the heavily doped TJs regions and therefore they do not affect the tunneling for both bottom and top TJs. As a result the tunneling properties of both TJ are the same for both tunneling directions as shown in Supplementary Materials in Fig. S1.

(lines 230-240)

The turn-on voltage of a BD LED is about 2V higher than that of a conventional blue III-N LEDs. This is caused mainly by the additional voltage drop across the reverse biased TJ that operates in the tunneling mode. We calculated the tunneling current for both top and bottom TJs (presented in the Supplementary Materials in Fig. S3) with our recently developed tunneling model¹⁷. We estimate that the turn-on voltages on BD LED increases by 1.2 V due to voltage drop across the TJ that is polarized reverse. Moreover, calculations show that both TJs have identical tunneling IV characteristics. The remaining difference in voltage may be due to the forward bias TJ. The turn-on voltage of a forward biased p-n junction depends on the energy gap of the semiconductor from which the junction is made. Therefore, a forward biased p-n junction made of $\text{In}_{0.02}\text{Ga}_{0.98}\text{N}$ will require a higher voltage than a conventional LED with an $\text{In}_{0.17}\text{Ga}_{0.83}\text{N}$ QW. Additional energetic barriers formed at the interfaces between the p-type region and the undoped $\text{In}_{0.08}\text{Ga}_{0.92}\text{N}$ layer may be source of asymmetry in the IV characteristic of BD LED.

Supplementary Materials:

In Fig. S1 we present the calculated tunneling currents as a function of voltage for bottom and top TJ used in examined BD LEDs. They were calculated using a previously developed model that takes into account the spontaneous and piezoelectric charges¹. In the simulations, the structural parameters of the TJs were set to be identical to those of the top and bottom TJs in the BD LED. As can be seen the calculated tunneling currents are identical for both TJs. The current-voltage characteristics of both TJs is the same, because piezoelectric sheet charges that occur at $\text{GaN}/\text{In}_{0.02}\text{Ga}_{0.98}\text{N}$ and $\text{In}_{0.02}\text{Ga}_{0.98}\text{N}/\text{In}_{0.08}\text{Ga}_{0.92}\text{N}$ interfaces are located outside of the junction region (see Fig. 2 in main article). Therefore, these charges do not affect tunneling, which will depend only on doping and bandgap, which are identical in top and bottom TJ.

5. Can the authors add one or more polarization and electric field direction schematics or band diagrams?

We have split the original Fig. 2 into two separate figures that are presented below as Fig. R7 and Fig. R8 (Fig. 2 and Fig. 3 in the revised manuscript). In Fig. R7 we added the graphs of electric field (F), polarization sheet charges (σ) and electron and hole concentrations (n , p). The added charts are described in detail in the revised manuscript. Moreover, we added Fig. R9 to supplementary materials (as Fig. S2), in which we present the electric field, electron and hole concentrations (n , p), electron and hole current density diagrams of BD LED active region under positive and negative bias, respectively. In addition, in Fig. R10 we present two band diagrams of stacked BD LEDs operating under negative and positive bias, in order to explain the carrier flow in BD LED stacks. This is needed to clarify the idea of BD LED stacking process. It is Fig. S3 in supplementary materials.

Fig. R7 (Fig. 2 in revised manuscript) (a) Schematic structure of a single BD LED. The corresponding (b) band diagram, (c) electric field (F) and polarization sheet charges (σ), (d) electron and hole carrier concentration (n , p) present in the BD LED under zero bias.

Fig. R8 (Fig. 3 in revised manuscript) (a, d) Power supply schemes, (b, e) band diagrams and (c, f) magnification of the active region of the BD LED under positive and negative biases, respectively. Arrows present the direction of electron and hole currents. The band diagrams were calculated for a current density of 1 and -1 kA/cm².

Fig. R9 (Fig. S2 in Supplementary Materials) (a, b) Power supply schemes and band diagram of the BD LED active region under positive (1 kA/cm²) and negative (-1 kA/cm²) biases, respectively. (e, f) Electric fields, (g, h) electron, hole current densities and (i, j) electron, hole carrier concentration present in the active region of BD LEDs under positive and negative biases, respectively. Data were obtained by simulation using the 1D-DDCC solver²⁻⁴.

Fig. R10 (Fig. S3 in Supplementary Materials) (a, c) Power supply schemes and (b, d) band diagrams of stacks of two BD LEDs under positive and negative bias regime, respectively, calculated for current density of 1 kA/cm^2 and -1 kA/cm^2 . Arrows present the direction of electron and hole currents.

Changes in manuscript:

(lines 130-143)

We study the BD LED properties by analyzing its band diagrams. A simplified structure of unbiased single BD LED is presented in Fig. 2a, whereas Figs. 2b, 2c and 2d show its band diagram, electric field (F) together with polarization sheet charges (σ) and electron and hole concentrations (n , p), respectively. The active region of BD LED is located between two p-type regions, rather than between a p-type and a n-type, like in conventional LEDs. Although the epitaxial structure of the BD LED is symmetrical with respect to the QW, its band structure presented in Fig. 2b is not. This is a consequence of polarization charges (both spontaneous and piezoelectric) characteristic to growth performed on Ga-polar side (0001) GaN

substrate³⁹. Polarization sheet charges (σ), which appear at the interfaces where the In composition is changed are presented in Fig. 2c. The greatest influence of the polarization charges on to band arrangement is found in the active region. The Fermi-level on the left-hand side of the QW is pinned to the p-type by holes accumulated at $\text{In}_{0.08}\text{Ga}_{0.92}\text{N}/\text{In}_{0.17}\text{Ga}_{0.83}\text{N}$ interface with negative polarization charges (see Fig. 2c and 2d). In contrast, no carriers accumulate on the right-hand side of the QW at the $\text{In}_{0.17}\text{Ga}_{0.83}\text{N}/\text{In}_{0.08}\text{Ga}_{0.92}\text{N}$ interface. However, due to the positive polarization charges there is an electric field (F) of $-1.2 \cdot 10^6 \text{ V} \cdot \text{cm}^{-1}$ present in the QW.

(lines 167-182)

The biasing convention of the BD LEDs is as follows: the BD LED is positively biased, when we apply a positive potential to the top electrical contact, and negatively, when we apply a negative potential, like it is schematically shown in Figs. 3a and 3d, respectively. Figs. 3b and 3e show the band diagrams of single BD LED under positive and under negative biases, respectively. Figs. 3c and 3f show the magnification of the band diagrams in the QW region. The band diagrams were calculated for current density of $1 \text{ kA}/\text{cm}^2$ and $-1 \text{ kA}/\text{cm}^2$. The bottom TJ is located on the left-hand side, while the top TJ on the right-hand side of the active region. Both TJs are able to operate in two modes: carrier tunneling when reverse biased, or carrier drift-diffusion transport when forward biased. The biasing conditions of top or bottom TJ is imposed by the bias of the whole BD LED. Thanks to this property, the carriers are able to reach the same QW in both positive and negative biases of a single BD LED.

(lines 204-219)

Utilization of two TJs, one on each side of the active region, makes the BD LED structure symmetrical. However, the growth on Ga-polar (0001) GaN substrate introduces an asymmetry in the QW, due to the arrangement of the built-in spontaneous and piezoelectric sheet charges³⁹. The barrier for electron escape from the QW towards the right-hand side is lower than barrier for escape towards the left-hand side as can be seen in Figs. 3c and 3f. When the positive bias is applied and the electrons are injected into the QW from the left-hand side, the low barrier can lead to electron escape from the QW to the right-hand side. Therefore, taking into account the direction of the carrier injection, the band alignment in the active region is more favorable for negative bias (see Figs. 3e and 3f) and leads to high injection efficiency. In the case of positively biased BD LED, the barrier for electron escape from the QW is small (Fig. 3c) and the resulting injection efficiency is low. The electrons, which overshoot the QW under positive bias, can give rise to parasitic recombination with holes outside of the QW. We discuss the effect of built-in electric field on the positively and negatively biased active region of the BD LED in more details in the Supplementary Materials in Fig. S2.

Furthermore, several BD LED can be stacked together in one epitaxial process in order to obtain higher optical power or multi-color emission. The principle of operation of stack of two or more BD LEDs is discussed in the Supplementary Materials in Fig. S3.

Supplementary Materials:

In Fig. S2 we present the calculated properties of bidirectional light emitting diode (BD LED) under positive and negative bias. The simulation was performed using 1D-DDCC solver²⁻⁴ at a current density of $-1 \text{ kA}/\text{cm}^2$ and $1 \text{ kA}/\text{cm}^2$, respectively. Figs. S2a and S2b show the schematics of BD LED with applied positive and negative voltage, respectively. Figs. S2c and S2d present the band structure of the active region. The

dashed arrow shows the overshoot of electrons due to unfavorable arrangement of the electric field in InGaN quantum well (QW) under positive bias (see Fig. S2c). The difference between the performance of a BD LED at positive and negative bias is significant in the active region. Although the arrangement of the conduction and valence bands and consequently the electric field in the quantum well for both structures are almost the same (see Figs. S2c-d and Figs. S2e-f), it is the crystallographic direction, from which carriers are delivered, that matters. Due to built-in electric fields that are characteristic of growth on Ga-polar GaN substrates, the barrier for electrons in the conduction band below the QW is significantly higher than above it, while in the valence band the barrier for holes is higher above the QW. Therefore, if electrons are supplied along the growth direction and holes from the top, they both face energy barriers preventing carrier escape. The resulting injection efficiency is almost 100% and all the current goes to carrier recombination in the QW. This is the case of negatively biased BD LED. On the other hand, when BD LED is positively biased, the electrons, due to low barrier are prone to escape from the QW. Indeed, this can be observed when analyzing the calculated electron and hole currents, which are presented in Figs. 2g and 2h. We see that in case of positively biased BD LED, part of the electron current overshoots the QW. As expected, in case of biased BD LED, all there is no electron nor hole overshoot and the carriers recombine in the QW. In Figs. 2i and 2j we present carrier concentrations. As can be seen, the two cases, positive and negative biases, do not differ much. The only difference is in the increased hole concentration right above the QW in case of the positive bias (see Fig. 2i).

These band diagram and current flow simulations qualitatively explain the difference in optical power between positive and negative power supply of the BD LED in experiment. In addition, they support the hypothesis of electron overflow, which, we claim, is the main reason for the parasitic peak in electroluminescence that appears at $\lambda=420-430$ nm for positively biased BD LED.

Fig. S3 presents the band diagram of a stack of two BD LEDs fabricated in a single epitaxial process. The carrier flow inside each section from the stack is the same as in a single BD LED. However, in this case, the current flow between each individual BD LEDs in the stack is important. For both positive and negative bias it is only the electron current.

6. I do not understand what is opening voltage, is it “turn on” voltage’?

The reviewer is correct, the term "opening voltage" is wrong. Of course, we meant turn-on voltage. Appropriate changes have been made in the manuscript.

Changes in manuscript:

(lines 224-228)

For single BD LED the turn-on voltages are approximately 5 V and -5.5 V, while the operating voltage, for current densities of 1 kA/cm^2 and -1 kA/cm^2 are 6.9 V and -8 V.

7. Can the authors provide any absolute number of light output power instead of an arbitrary unit?

The measurement presented in the manuscript were performed on-wafer with a power meter placed in a manner not optimized for light collection. Therefore, we had to redo the optical power measurements. We cleaved the device, mounted it and measured with an integrating sphere. The results are presented below on Fig. R11. Single BD LED reaches 2.5 mW and 0.24 mW of optical power, whereas

stack of two BD LEDs reaches 4.6 mW and 0.36 mW at -1 kA/cm^2 and 1 kA/cm^2 , respectively. The present results determine an EQE for a single BD LED equal to 0.9% and 0.09% at -100 mA and 100 mA , respectively.

We would like to note that the devices are not optimized for light extraction, so the results cannot be compared one-to-one with commercial LEDs. With processing that we used in this work, a lot of light is lost due to absorption on thick metal contacts and there is no encapsulations of the devices. We expect that after optimization for light extraction, we can achieve several times higher optical power. These steps are required to make BD LED comparable to well-developed conventional LEDs.

Fig. R11 (Fig. 4 in revised manuscript) (a) Current-voltage and (b) optical power characteristics of single and stack of two BD LEDs. Optical spectra of (c-d) single and (e-f) stack of two BD LEDs under negative and positive bias measured in steps of 0.2 kA/cm^2 up to 1 kA/cm^2 , respectively. Insets show real color pictures showing emission at a current density of $\pm 1 \text{ kA/cm}^2$ obtained from devices with a size of $100 \times 100 \mu\text{m}^2$.

Changes in manuscript:

(lines 248-259)

The dependence of optical power on current density is presented in Fig. 4b. Single BD LED reaches 2.5 mW and 0.24 mW of optical power, whereas stack of two BD LEDs reaches 4.6 mW and 0.36 mW at -1 kA/cm² and 1 kA/cm², respectively. The external quantum efficiency (EQE) is 0.9% and 0.09% at -1 kA/cm² and 1 kA/cm² for single BD LED. As already mentioned, the injection efficiency is higher for BD LED under negative than positive bias. We consider this to be the main reason of 10-times higher optical power of BD LED under the negative bias.

We would like to note that this BD LED has not yet been optimized for light extraction, so the results cannot be compared one-to-one with commercial LEDs. With processing that we used in this work, a lot of light is absorbed in thick metal contacts which cover the majority of the surface. Additionally, there is no encapsulations of the devices, which would enhance the extraction of light. We expect that after optimizing the light extraction from BD LEDs, we can achieve several times higher optical power. These steps are required to make BD LED comparable to well-developed conventional LEDs.

8. In lines 141-142, “in the case of the single BD LED, the collected light power grows approximately linearly with the positive and negative supply currents”. I do not understand this statement. Since the TJ works better in reverse bias, this is why the LED where the TJ is reverse biased works better (10 times higher luminous power). Overall, the paragraph is hard to follow (lines 141-148).

As we discussed in the answer to your 2nd comment we made use of dual behavior of TJs that are: carrier tunneling, when TJ is under reverse bias and carrier drift-diffusion, when TJ is under forward bias. Moreover, as we presented in Fig. R5 both top and bottom TJs have the same tunneling current characteristic, so it is not the process of tunneling through the TJs that determines the difference in light emission from the BD LED in positive and negative biases.

The real reason why we achieve a 10-times higher electroluminescence when the BD LED is under negative bias than when it is under positive bias is the existence of built-in electric fields in the active region of diode that causes the difference in the injection efficiency of carriers depending on the direction of current supply. Indeed, when the LED is under positive bias, electrons that reach the QW from the left-hand side may easily overflow the QW, because the right-hand side barrier of the QW is low. On the other hand, when BD LED is powered negatively, the left-hand side barrier of the QW is higher and acts as an efficient electron blocking layer. This situation is well depicted in Fig. R12, which we added to Supplementary Materials as Fig. S2.

The paragraph (lines 141-148) has been changed.

Fig. R12 (Fig. S2 in Supplementary Materials) (a, b) Power supply schemes and band diagram of the BD LED active region under positive (1 kA/cm²) and negative (-1 kA/cm²) biases, respectively. (e, f) Electric fields, (g, h) electron, hole current densities and (i, j) electron, hole carrier concentration present in the active region of BD LEDs under positive and negative biases, respectively. Data were obtained by simulation using the 1D-DDCC solver²⁻⁴.

Changes in manuscript:

(lines 248-253)

The dependence of optical power on current density is presented in Fig. 4b. Single BD LED reaches 2.5 mW and 0.24 mW of optical power, whereas stack of two BD LEDs reaches 4.6 mW and 0.36 mW at -1 kA/cm² and 1 kA/cm², respectively. The external quantum efficiency (EQE) is 0.9% and 0.09% at -1 kA/cm² and 1 kA/cm² for single BD LED. As already mentioned, the injection efficiency is higher for BD LED under negative than positive bias. We consider this to be the main reason of 10-times higher optical power of BD LED under the negative bias.

(lines 260-269)

In the case of stack of two BD LEDs the luminous power is roughly 1.8 times higher than in the case of single BD LED. This result shows that by stacking BD LEDs, one can increase the optical power obtained with the same supply current.

Supplementary Materials:

In Fig. S2 we present the calculated properties of bidirectional light emitting diode (BD LED) under positive and negative bias. The simulation was performed using 1D-DDCC solver²⁻⁴ at a current density of -1 kA/cm^2 and 1 kA/cm^2 , respectively. Figs. S2a and S2b show the schematics of BD LED with applied positive and negative voltage, respectively. Figs. S2c and S2d present the band structure of the active region. The dashed arrow shows the overshoot of electrons due to unfavorable arrangement of the electric field in InGaN quantum well (QW) under positive bias (see Fig. S2c). The difference between the performance of a BD LED at positive and negative bias is significant in the active region. Although the arrangement of the conduction and valence bands and consequently the electric field in the quantum well for both structures are almost the same (see Figs. S2c-d and Figs. S2e-f), it is the crystallographic direction, from which carriers are delivered, that matters. Due to built-in electric fields that are characteristic of growth on Ga-polar GaN substrates, the barrier for electrons in the conduction band below the QW is significantly higher than above it, while in the valence band the barrier for holes is higher above the QW. Therefore, if electrons are supplied along the growth direction and holes from the top, they both face energy barriers preventing carrier escape. The resulting injection efficiency is almost 100% and all the current goes to carrier recombination in the QW. This is the case of negatively biased BD LED. On the other hand, when BD LED is positively biased, the electrons, due to low barrier are prone to escape from the QW. Indeed, this can be observed when analyzing the calculated electron and hole currents, which are presented in Figs. S2g and S2h. We see that in case of positively biased BD LED, part of the electron current overshoots the QW. As expected, in case of biased BD LED, all there is no electron nor hole overshoot and the carriers recombine in the QW. In Figs. S2i and S2j we present carrier concentrations. As can be seen, the two cases, positive and negative biases, do not differ much. The only difference is in the increased hole concentration right above the QW in case of the positive bias (see Fig. S2i).

9. It is unclear why the BD LED stays turned on just more than half the total pulse period? I think the BD LED is much brighter in one half (when the TJ is reverse biased and current tunneling happens), whereas in the other half, it should be much dimmer as TJ is forward biased and no tunneling occurs. It should be much brighter if it works with one LED but will not be bidirectional.

We intended to make devices which are bidirectional i.e. the devices should stay turned on in both positive and negative half-cycles of sinusoidal voltages. This is the main difference that distinguishes our BD LED from conventional LEDs. Under the positive half-cycle of the BD LED, the bottom TJ is forward biased, while the top TJ is reverse biased. On the other hand, under the negative half-cycle of the BD LED, the top TJ is forward biased, while the bottom TJ is reverse biased. Thanks to this design, the carriers are able to reach the same QW in both positive and negative bias of single BD LED and consequently, the whole device stays turned on for more than half of the total pulse period.

We have made changes in the biasing convention and BD LED operation under positive and negative bias is shown in Fig. R13. The reason why under the positive voltage half-cycle the BD LED is much dimmer is the lower injection efficiency in this biasing regime. This was discussed in detail above in the answer to your 8th comment.

Fig. R13 (Fig. 3 in revised manuscript) (a, d) Power supply schemes, (b, e) band diagrams and (c, f) magnification of the active region of the BD LED under positive and negative biases, respectively. Arrows present the direction of electron and hole currents. The band diagrams were calculated for a current density of 1 and -1 kA/cm².

10. The statement in lines 231-232, “In fact, the positive and negative bias conditions of the BD LED can be directly compared to III-N LEDs grown on Ga-polar and N-polar GaN substrates, respectively.” This statement is not valid. It cannot be the same because the polarization directions and spontaneous and polarizations fields will differ for each layer. This statement should be clarified.

We agree with reviewer, that we should be more precise in this statement. Indeed, the active region in BD LEDs is located between two p-type regions, rather than between a p-type and an n-type, as in conventional LEDs, which significantly alters their band structure for unbiased condition. However, for biased structures, if one considers the arrangement of built-in electric fields and the direction of carrier injection into the QW, the active region of positively biased BD LED resembles conventional LEDs grown on Ga-polar substrate, while a negatively biased BD LED resembles conventional LEDs grown on N-polar substrate, respectively. In these terms the negatively biased BD LED works similarly to an “inverted” LED described in [7]. Furthermore, experimental results are consistent with this statement: negatively biased BD LEDs is characterized with a higher injection efficiency, such as LEDs grown on N-polar substrate. On

the other hand, positively biased BD LED suffer from low injection efficiency, like conventional LEDs without an electron blocking layer grown on Ga-polar substrate.

We change this invalid statement and made other minorities changes in paragraph *Influence of the QW polarization field on carrier injection* (lines 344-417).

Changes in manuscript:

(lines 360-366)

When the BD LED structure is biased, then it is interesting to consider the direction of carrier injection into the QW with respect to the arrangement of built-in electric fields. In these terms, for positive bias, the BD LED resembles conventional LEDs grown on Ga-polar substrate, while for negative bias it resembles LEDs grown on N-polar substrate. In such LEDs, the differences in the injection efficiencies for structures with various polarities have been widely discussed^{19-22,46-48}.

11. How many LEDs can be stacked by TJ using planar growth process? The overall voltage of the stacked device is still low (maybe there are some leakage currents).

This is a very good point to emphasize that the device stacking process can be carried out multiple times. We do not see any limitation on the number of LEDs, as the crystal quality of the structure remains very good [8,9].

Indeed, stack of two BD LEDs operates at a slightly lower voltage than expected, when multiplying the voltage of the single BD LED by a factor of two. The voltage on stack of two BD LEDs should be slightly lower twice the value of voltage on a single device, because the resistances of substrate and metal contacts are common in both cases. Only the resistance of the diodes and TJs is doubled.

Changes in manuscript:

(lines 241-247)

In case of the stack of two BD LEDs, the turn-on voltages are 9.4 V and -11.2 V, for positive and negative biases, respectively. The operating voltage, for current densities of 1 kA/cm² and -1 kA/cm², are 12.5 V and -13.3 V, respectively. Stack of two BD LEDs operates at a slightly lower voltage than expected, when multiplying the voltage of the single BD LED by a factor of two. Indeed, the voltage on stack of two BD LEDs should be slightly lower than twice the value of voltage on a single device, because the resistances of substrate and metal contacts are common in both cases. Only the resistance of the diodes and TJs is doubled.

12. Can the authors add more details about the growth of TJ as well as LED? It would be interesting to see the growth rate/duration for the stacked structures.

The growth rate of GaN layers was 0.36 $\mu\text{m/h}$, the QW was grown with the rate of 0.76 $\mu\text{m/h}$, whereas other InGaN layer with 0.85 $\mu\text{m/h}$. The approximate growth time of a single BD LED structure, together with the time required for the change in growth temperature between GaN and InGaN layers, was 2 hours. Therefore, the duration of the growth of a stack of n BD LED, will be equal to n times 2 hours. However it can be reduced in the next generation of BD LEDs stacks, by elimination of breaks needed to

switch from InGaN to GaN growth conditions. It can be done by changing the 200 nm thick GaN:Si interconnecting layer to thinner $\text{In}_{0.01}\text{Ga}_{0.99}\text{N}:\text{Si}$.

We added information about growth process in Materials and Methods.

Changes in manuscript:

(lines 515-530)

In case The structures were grown in metal-rich conditions using plasma-assisted molecular beam epitaxy (PAMBE) on the Ga-polar (0001) side of n-type conductive ($n \approx 10^{18} \text{ cm}^{-3}$) commercially available GaN bulk crystals obtained using hydride vapor phase epitaxy. The growth was performed in a VG Semicon V90 reactor. GaN layers were grown at 730 °C with a growth rate of 0.36 $\mu\text{m}/\text{h}$. InGaN layers were grown at 650 °C with growth rates of 0.76 $\mu\text{m}/\text{h}$ and 0.85 $\mu\text{m}/\text{h}$ for QW and other InGaN layers, respectively. The growth temperature was controlled in situ by laser reflectometry⁵³. Growth interrupts, which are visible in TEM presented in Fig. 1c as dark horizontal lines, were performed to control metal coverage during InGaN growth. The indium content of the InGaN layers was controlled by adjusting the gallium to nitrogen ratio according to the model presented in the paper of Turski et al.⁵⁴. The approximate growth duration of a single BD LED structure, including the time necessary to stabilize growth condition in the reactor, was 2 hours. Therefore, the time needed to grow a stack of n BD LEDs will be n times 2 hours. Details of PAMBE growth can be found elsewhere⁵².

13. Why were asymmetric positive 100 A/cm² and negative -60 A/cm² applied for the time-resolved electroluminescence measurement?

In this experiment, we set a symmetrical supply voltage, which corresponded to +100 A/cm² for the positive pulse and -60 A/cm² for the negative pulse.

Changes in manuscript:

(lines 309-311)

The single BD LED device was biased alternately with symmetrical supply voltage of 6.1 V and -6.1 V that corresponded to positive +100 A/cm² and negative -60 A/cm² pulses, both 5 μs long.

14. Overall, the writing can be improved; the manuscript is hard to read.

We have improved the language and grammar of the revised manuscript. We have made minor corrections throughout the article in track changes. The description of the effect of built-in electric fields on BD LED operation has been moved and modified from *Discussion* (lines 345-359) to *BD LED operating principle* (lines 204-216).

Reviewer #3 (Remarks to the Author):

This manuscript studies a bidirectional vertical InGaN LED with two TJs which can be AC driven which is interesting. Though there are several concerns from this study:

1. The TEM image in Fig. 1(c) is unclear. The contrast from TJs and active region is not good enough. It will be better if an improved TEM can be provided.

Indeed, during the previous STEM examination the contrast in the STEM microscope was not the finest. As suggested, we repeated the STEM images, obtaining a higher contrast, as shown in Fig. R14.

Fig. R14 (Fig. 1 in revised manuscript) Schematic structure of (a) single and (b) stack of two bidirectional light emitting diodes (BD LEDs). (c) STEM cross-section image of the stack of two BD LEDs. Visible dark horizontal lines are growth interrupts.

Changes in manuscript:

(lines 524-526)

Growth interrupts, which are visible in TEM presented in Fig. 1c as dark horizontal lines, were performed to control metal coverage during InGaN growth.

2. Also, there has been studies on the use of bottom and top TJs - how would this work compare to those existing works? What are the novelties?

LEDs with single TJ (bottom or top) resemble conventional LED structure – QW is located inside p-n junction, i.e. lies between p and n-type layers. TJs in those structures are used to change the type of conductivity of majority carriers e.g. from hole in p-type to electron in n-type. Such device with single TJ can operate only in one biasing regime.

Our BD LED structures have both the bottom and top TJs with a QW in between. The novelty is that such a design can operate in both positive and negative bias.

The unique design of our BD LED requires the QW to be surrounded by p-type region from both sides. This special construction of the BD LED, and the fact that electrons can be easily transported through the p-type region (as we discuss in manuscript), allows for carrier injection into the same QW under positive and negative biases.

Changes in manuscript:

(lines 78-92)

It should be emphasized here that during the last decade III-N TJs have reached maturity and a high tunneling current under low operating voltage can be achieved^{8,16-18}. TJ are used to stack several LEDs or laser diodes (LDs) for multiplying the optical power or obtaining multi wavelength optical spectrum²³⁻²⁷, to control current flow in micro LEDs²⁸⁻³¹, in distributed-feedback LDs³² or in tunneling field-effect transistors³³. Additionally, TJs can be used to invert the direction of current flow with respect to the arrangement of piezoelectric fields in the active region¹⁹⁻²². However, LEDs with only a single TJ (bottom or top) still resemble conventional LED structures, in which the QW is located inside the p-n junction of a LED. Such devices can only operate in one bias regime, i.e. the LED or LD is forward biased and TJ is reverse biased^{8,17-32}. The unique design of our BD LED, in which a single QW is surrounded by two TJs, one from each side, allows for operation in both bias regime.

3. How are the results in Fig. 2 generated? Please describe. How are the TJs designed for this LED? For example, what are the roles of thickness, doping, etc.?

The schemes shown in Fig. 2 was generated using 1-dimensional drift-diffusion charge control solver (1D DDCC) from prof. Y.-R. Wu with a mesh of 0.1 nm [10-12]. The piezo + spontaneous polarization constants were taken from the work of F. Bernardini et al [13]. The band diagram of unbiased BD LED structure was calculated directly in 1D-DDCC, whereas the biased ones were split in two parts. For positively biased BD LED the first part constitutes a forward biased bottom TJ and the active region with the p-type layer above it. Within 1D-DDCC solver the top TJ is not simulated, because there is tunneling effect is not implemented in 1D-DDCC solver. The second part consists of the upper p-type region and the top TJ that was reverse biased. Tunneling current was calculated using our model published in Ref [6]. These two band diagrams were later merged in the middle of the p-type region, in which the bands are flat. For negative bias the splitting of BD LED structure was analogous – in the middle of the bottom p-type region. We have split the original Fig. 2 into two separate figures, Fig. 2 and Fig. 3 in the revised manuscript.

As for the TJ design, we intended to obtain transparent homojunctions with identical tunneling current characteristic. In the presented BD LEDs TJs are composed from $\text{In}_{0.02}\text{Ga}_{0.98}\text{N}$ heavily doped with Si (20 nm) and Mg (10 nm) both at the level of $1 \times 10^{20} \text{ cm}^{-3}$. This 30 nm heavily doped region is larger than the depletion width of TJ, so the electric field (F) at the junction is only the field originating from the ionized dopants, as it is presented on Fig. R15c.

Spontaneous and piezoelectric charges, that occur at $\text{GaN}/\text{In}_{0.02}\text{Ga}_{0.98}\text{N}$ and $\text{In}_{0.02}\text{Ga}_{0.98}\text{N}/\text{In}_{0.08}\text{Ga}_{0.92}\text{N}$ interfaces are located outside of the junction region (see Fig. R15c). Therefore, these charges do not influence the electric field of the TJ and have no impact on tunneling currents and tunneling efficiency for both bottom and top TJs. To validate our considerations, we calculated tunneling currents using a previously developed model that takes into account the spontaneous and piezoelectric charges [6]. Fig. R16 shows the tunneling current as a function of voltage for bottom and top TJ used in examined BD LEDs. As can be seen the calculated tunneling currents are identical for both TJs. We present this chart in the supplementary materials as Fig. 3.

Fig. R15 (Fig. 2 in revised manuscript) (a) Schematic structure of a single BD LED. The corresponding (b) band diagram, (c) electric field (F) and polarization sheet charges (σ), (d) electron and hole carrier concentration (n , p) present in the BD LED under zero bias.

Fig. R16 (Fig. S1 in supplementary materials) Calculated tunneling current for bottom and top TJs of BD LED.

Changes in manuscript:

(lines 144-151)

The TJs dedicated for BD LEDs are designed to have the same tunneling characteristics for the top and bottom configurations and to be transparent to blue light. Therefore, we use $\text{In}_{0.02}\text{Ga}_{0.98}\text{N}$ homojunctions in which the heavily doped region is much larger than the depletion width of the junction. Hence, the electric field (F) in both TJs originates only from the ionized dopants, as it is presented in Fig. 2c. Polarization sheet charges, that occur at $\text{GaN}/\text{In}_{0.02}\text{Ga}_{0.98}\text{N}$ and $\text{In}_{0.02}\text{Ga}_{0.98}\text{N}/\text{In}_{0.08}\text{Ga}_{0.92}\text{N}$ interfaces are

located outside of depletion width of the heavily doped TJs regions and therefore they do not affect the tunneling for both bottom and top TJs. As a result the tunneling properties of both TJ are the same for both tunneling directions as shown in Supplementary Materials in Fig. S1.

(lines 177-178)

The band diagrams were calculated for current density of 1 kA/cm² and -1 kA/cm².

(lines 547-557)

The band diagrams shown in Figs. 2 and 3 were generated using 1-dimensional drift-diffusion charge control solver (1D-DDCC) by Y.-R. Wu⁴⁹⁻⁵¹. The mesh was set to 0.1 nm, whereas electron and holes mobilities were 300 cm²V⁻¹s and 10 cm²V⁻¹s, respectively. The spontaneous and piezoelectric polarization constants were taken from the paper of F. Bernardini et al.³⁹. The band diagram of unbiased BD LED structure was calculated directly in 1D-DDCC, whereas the biased band diagrams were split in two parts. For positively biased BD LED the first part constituted a forward biased bottom TJ and the active region with the p-type layer above it. Within 1D-DDCC solver the top TJ is not simulated, because tunneling is not implemented in 1D-DDCC solver. The second part consists of the upper p-type region and the top TJ that was reverse biased. Tunneling current was calculated using our model published in Ref¹⁷. These two band diagrams were later merged in the middle of the p-type region, where the bands are flat. For negative bias the splitting of BD LED structure was analogous – in the middle of the bottom p-type region.

Supplementary Materials:

In Fig. S1 we present the calculated tunneling currents as a function of voltage for bottom and top TJ used in examined BD LEDs. They were calculated using a previously developed model that takes into account the spontaneous and piezoelectric charges¹. In the simulations, the structural parameters of the TJs were set to be identical to those of the top and bottom TJs in the BD LED. As can be seen the calculated tunneling currents are identical for both TJs. The current-voltage characteristics of both TJs is the same, because piezoelectric sheet charges that occur at GaN/In_{0.02}Ga_{0.98}N and In_{0.02}Ga_{0.98}N/In_{0.08}Ga_{0.92}N interfaces are located outside of the junction region (see Fig. 2 in main article). Therefore, these charges do not affect tunneling, which will depend only on doping and bandgap, which are identical in top and bottom TJ.

4. It seems like the turn on voltage is very large from Fig. 3. Please explain.

The turn-on voltage of a BD LED is about 2V higher than the turn-on voltage of a standard blue LED. This is mainly due to the additional voltage drop across the reverse biased TJ, which increases the turn-on voltage by 1.2 V. We expect that the voltage drop on TJ can be lowered by increasing the doping level in the TJ or the InGaN composition of the TJ. On Fig. R17 we present the calculated tunneling currents through TJs as a function of voltage, for actual and higher doping levels. One can see that by increasing concentration of both type dopants to the level of 2.5x10²⁰ cm⁻³ the turn-on voltage can be reduced to 0.3 V. Such high doping level is possible – e.g. for n-type dopants, we already reported doping levels higher than 5x10²⁰ cm⁻³ for In_{0.02}Ga_{0.98}N:Ge layers [14].

The remaining difference in voltage may be due to the need to forward bias the second of the tunnel junctions to inject electrons into the QW. The turn-on voltage of a positively biased p-n junction depends on the energy gap of the semiconductor from which the junction is made. Therefore, a positively biased p-n junction made of In_{0.02}Ga_{0.98}N will require a higher voltage than a LED with a In_{0.17}Ga_{0.83}N QW.

In addition, the higher voltage may be due to additional barriers formed at the interface between the p-type region and the undoped $\text{In}_{0.08}\text{Ga}_{0.92}\text{N}$ barriers.

Fig. R17 Dependence of tunneling current on voltage for $\text{In}_{0.02}\text{Ga}_{0.98}\text{N}$ homojunction with symmetrical p- and n-type doping.

Changes in manuscript:

(lines 230-240)

The turn-on voltage of a BD LED is about 2V higher than that of a conventional blue III-N LEDs. This is caused mainly by the additional voltage drop across the reverse biased TJ that operates in the tunneling mode. We calculated the tunneling current for both top and bottom TJs (presented in the Supplementary Materials in Fig. S3) with our recently developed tunneling model¹⁷. We estimate that the turn-on voltages on BD LED increases by 1.2 V due to voltage drop across the TJ that is polarized reverse. Moreover, calculations show that both TJs have identical tunneling IV characteristics. The remaining difference in voltage may be due to the forward bias TJ. The turn-on voltage of a forward biased p-n junction depends on the energy gap of the semiconductor from which the junction is made. Therefore, a forward biased p-n junction made of $\text{In}_{0.02}\text{Ga}_{0.98}\text{N}$ will require a higher voltage than a conventional LED with an $\text{In}_{0.17}\text{Ga}_{0.83}\text{N}$ QW. Additional energetic barriers formed at the interfaces between the p-type region and the undoped $\text{In}_{0.08}\text{Ga}_{0.92}\text{N}$ layer may be source of asymmetry in the IV characteristic of BD LED.

5. Please also explain the dual peaks from Figs. 3(d) and 3(f).

The second peak present in the spectra in the figures comes from the recombination of electrons that overflow over the QW in the $\text{In}_{0.02}\text{Ga}_{0.98}\text{N}:\text{Mg}$ layer. In the case of conventional LEDs grown on Ga-polar structures, low injection efficiency and electron overflow above the QW is a commonly known issue. This problem is overcome by introduction of an electron blocking layer (EBL) on the p-side right after the QW [15]. Heavily doped AlGa_N:Mg layer is most commonly used as the EBL, which due to its higher bandgap, forms an energetic barrier for electrons. When the EBL is not present or is not working properly electrons that pass over the QW recombine in the p-type region. This leads to the parasitic recombination in the spectral range of $\lambda=420\text{-}430 \text{ nm}$ [16,17]. In literature this peak is often identified as transition from the unknown deep donor down to Mg acceptor level [18,19].

In the presented devices we did not use the EBL in order to keep the design symmetrical. Therefore, we observe the electron overflow and luminescence from p-type as in case of conventional LEDs without EBL. We plan to make BD LEDs with an EBL in the future.

We discuss the origin of this parasitic peak more detailed in lines 381-392 of revised manuscript.

Changes in manuscript:

(lines 273-275)

We attribute this parasitic peak to the recombination of electrons that overflow the active region, with holes in the $In_{0.02}Ga_{0.98}N:Mg$ layer (in between 235 nm and 270 nm in the band diagram shown in Fig. 3b)⁴⁴⁻⁴⁵.

6. Please describe the LED fabrication process in more details.

We have added a paragraph describing the LED fabrication process.

Changes in manuscript:

(lines 531-546)

Samples were processed into devices with dimensions of $100 \times 100 \mu m^2$ by standard lift-off photolithography process using LaserWriter. First, a negative photoresist NLOF 2020 was spin coated onto the samples and rows of $100 \times 100 \mu m^2$ squares were exposed. The mesa was etched using chlorine inductively coupled plasma reactive ion etching (ICP RIE) to a depth of 450 nm and 900 nm for single and stack of two BD LEDs, respectively. The photoresist was removed in dimethyl sulfoxide (DMSO) and a second photolithography process was carried out in which $90 \times 90 \mu m^2$ windows were opened at the top of each mesa, while the rest of the etched samples and sides of the mesa were protected. Next, a Ti/Al/Ni/Au (30/60/40/75 nm) metallic contact was sputtered on the top of sample, the photoresist was removed in dimethyl sulfoxide (DMSO) and then the metallic contact was annealed at 750 °C for 1 minute in an N_2 atmosphere. The bottom metallization was applied to the entire reverse N-side surface of the sample and was not annealed.

7. How does the power/EQE compare to other blue LEDs? Please provide a comparison discussion.

We measured the BD LEDs in integration sphere. The results are presented in Fig R18b. Single BD LED reaches 2.5 mW and 0.24 mW of optical power, whereas stack of two BD LEDs reaches 4.6 mW and 0.36 mW at -100 mA and 100 mA, respectively. Regarding the EQE they are 0.9% and 0.09% at -100 mA and 100 mA. The optical power obtained under the negative bias is comparable with optical power of conventional LED obtained in our laboratory. However, we would like to note that this BD LED has not yet been optimized for light extraction. With current processing, a lot of light is lost due to absorption on thick metal contacts. We expect that after optimization of light extraction [20], we can achieve several times higher optical power. These efforts are planned as future studies.

Fig. R18 (Fig. 4 in revised manuscript) (a) Current-voltage and (b) optical power characteristics of single and stack of two BD LEDs. Optical spectra of (c-d) single and (e-f) stack of two BD LEDs under negative and positive bias measured in steps of 0.2 kA/cm² up to 1 kA/cm², respectively. Insets show real color pictures showing emission at a current density of ± 1 kA/cm² obtained from devices with a size of 100x100 μm^2 .

Changes in manuscript:

(lines 248-259)

The dependence of optical power on current density is presented in Fig. 4b. Single BD LED reaches 2.5 mW and 0.24 mW of optical power, whereas stack of two BD LEDs reaches 4.6 mW and 0.36 mW at -1 kA/cm² and 1 kA/cm², respectively. The external quantum efficiency (EQE) is 0.9% and 0.09% at -1 kA/cm² and 1 kA/cm² for single BD LED. As already mentioned, the injection efficiency is higher for BD LED under negative than positive bias. We consider this to be the main reason of 10-times higher optical power of BD LED under the negative bias.

We would like to note that this BD LED has not yet been optimized for light extraction, so the results cannot be compared one-to-one with commercial LEDs. With processing that we used in this work, a lot of light is absorbed in thick metal contacts which cover the majority of the surface. Additionally, there is no encapsulations of the devices, which would enhance the extraction of light. We expect that after optimizing the light extraction from BD LEDs, we can achieve several times higher optical power. These steps are required to make BD LED comparable to well-developed conventional LEDs.

8. Please provide a discussion and figure on the internal field of the QWs for under forward bias vs. reverse bias. It's helpful to expand the physics of the device under AC condition.

According to reviewer suggestions we added the internal field with additional data. In Fig. R19 we present the electric field (F), electron and hole carrier concentrations (n , p), electron and hole current density diagrams of BD LED active region under positive and negative bias, respectively. We have included this graph in the supplementary materials as Fig. S2. As can be seen, the value of electric field in the QW does not differ between the BD LED biased positively and negatively. The electric field in the QW is partially screened by free carriers that occupy the QW. Although the electric field in the quantum well remains the same, we observe a difference in the optical power due to various injection efficiency for positive and negative bias.

Fig. R19 (Fig. S2 in Supplementary Materials) (a, b) Power supply schemes and band diagram of the BD LED active region under positive (1 kA/cm²) and negative (-1 kA/cm²) biases, respectively. (e, f) Electric fields, (g, h) electron, hole current densities and (i, j) electron, hole carrier concentration present in the active region of BD LEDs under positive and negative biases, respectively. Data were obtained by simulation using the 1D-DDCC solver²⁻⁴.

Changes in manuscript:

(lines 204-216)

Utilization of two TJs, one on each side of the active region, makes the BD LED structure symmetrical. However, the growth on Ga-polar (0001) GaN substrate introduces an asymmetry in the QW, due to the arrangement of the built-in spontaneous and piezoelectric sheet charges³⁹. The barrier for electron escape from the QW towards the right-hand side is lower than barrier for escape towards the left-hand side as can be seen in Figs. 3c and 3f. When the positive bias is applied and the electrons are injected into the QW from the left-hand side, the low barrier can lead to electron escape from the QW to the right-hand side. Therefore, taking into account the direction of the carrier injection, the band alignment in the active region is more favorable for negative bias (see Figs. 3e and 3f) and leads to high injection efficiency. In the case of positively biased BD LED, the barrier for electron escape from the QW is small (Fig. 3c) and the resulting injection efficiency is low. The electrons, which overshoot the QW under positive bias, can give rise to parasitic recombination with holes outside of the QW. We discuss the effect of built-in electric field on the positively and negatively biased active region of the BD LED in more details in the Supplementary Materials in Fig. S2.

Supplementary Materials:

In Fig. S2 we present the calculated properties of bidirectional light emitting diode (BD LED) under positive and negative bias. The simulation was performed using 1D-DDCC solver²⁻⁴ at a current density of -1 kA/cm² and 1 kA/cm², respectively. Figs. S2a and S2b show the schematics of BD LED with applied positive and negative voltage, respectively. Figs. S2c and S2d present the band structure of the active region. The dashed arrow shows the overshoot of electrons due to unfavorable arrangement of the electric field in InGaN quantum well (QW) under positive bias (see Fig. S2c). The difference between the performance of a BD LED at positive and negative bias is significant in the active region. Although the arrangement of the conduction and valence bands and consequently the electric field in the quantum well for both structures are almost the same (see Figs. S2c-d and Figs. S2e-f), it is the crystallographic direction, from which carriers are delivered, that matters. Due to built-in electric fields that are characteristic of growth on Ga-polar GaN substrates, the barrier for electrons in the conduction band below the QW is significantly higher than above it, while in the valence band the barrier for holes is higher above the QW. Therefore, if electrons are supplied along the growth direction and holes from the top, they both face energy barriers preventing carrier escape. The resulting injection efficiency is almost 100% and all the current goes to carrier recombination in the QW. This is the case of negatively biased BD LED. On the other hand, when BD LED is positively biased, the electrons, due to low barrier are prone to escape from the QW. Indeed, this can be observed when analyzing the calculated electron and hole currents, which are presented in Figs. S2g and S2h. We see that in case of positively biased BD LED, part of the electron current overshoots the QW. As expected, in case of biased BD LED, all there is no electron nor hole overshoot and the carriers recombine in the QW. In Figs. S2i and S2j we present carrier concentrations. As can be seen, the two cases, positive and negative biases, do not differ much. The only difference is in the increased hole concentration right above the QW in case of the positive bias (see Fig. S2i).

9. Other than using polarized substrate, are there any ways to use this structure on c-plane GaN substrate to achieve more symmetrical performances?

We agree with the reviewer that the simplest way to achieve symmetrical operation of a BD LED is to carry out the growth of the BD LED structure on a non-polar substrate. However, such substrates are much more expensive and the growth is more challenging. Therefore, it is important to indicate the direction of research that would enhance the optical power obtained for positive bias for devices grown on standard Ga-polar (0001) substrates.

To increase the injection efficiency for positively biased BD LED, we propose to use an electron blocking layer (EBL) above the active region. However, it is necessary to take a broader look at the band diagram of BD LEDs, instead of only implementing solutions that are widely used in standard III-N LEDs. We need to develop a suitable EBL that, first of all, blocks the overflow of electrons for a positively biased BD LED, but also does not disturb the injection of electrons into the active region under a negative bias. Therefore, the standard AlGaIn:Mg layer, which is typically used in standard III-N LEDs as an EBL, might not be the best solution since it will form an energy barrier for electron injection for the negatively biased BD LED. We think about using a heavily doped $\text{In}_{0.02}\text{Ga}_{0.98}\text{N}:\text{Mg}$ layer as an EBL. From preliminary drift-diffusion calculations, we obtained increased injection efficiency for the positively biased BD LED to 97% at $4\text{ kA}/\text{cm}^2$ using $\text{InGaIn}:\text{Mg}$ doped to the level of $6 \times 10^{19} \text{ cm}^{-3}$ as presented in Fig. R20. Importantly, such layer would not reduce the injection efficiency for the negatively biased BD LED. Implementation of a suitable EBL layer to balance optical power will be the subject of future research on BD LEDs.

Fig. R20 Calculated injection efficiency as a function of doping in 2% $\text{InGaIn}:\text{Mg}$ EBL.

Changes in manuscript:

(lines 393-411)

Considering the BD LEDs presented here as direct AC-driven light sources, efforts should be made to increase the optical power obtained under positive bias. It should be pointed out that the asymmetry in the operation of the BD LEDs presented here is solely due to the existence of built-in electric fields specific to III-N structures grown on polar substrates. Fabricating a BD LED on a non-polar GaN substrate or in a different material system, in which the issue of embedded polarization charges does not exist, should give symmetric light-current-voltage (LIV) characteristics, regardless of the current flow direction. However, non-polar GaN substrates are much more expensive, and the growth is more challenging. Therefore, it is important to indicate the direction of research that would enhance the optical power obtained for positively

biased BD LEDs grown on conventional Ga-polar (0001) substrates. It would be beneficial to develop an EBL that, first of all, blocks the overflow of electrons for a positively biased BD LED, but also does not disturb the injection of electrons into the active region under a negative bias. Therefore, the AlGaIn:Mg layer, which is typically used in conventional III-N LEDs as an EBL, might not be the best solution since it will form an energy barrier for electron injection for the negatively biased BD LED. Implementation of a suitable EBL layer to balance optical power will be the subject of future research on BD LEDs.

References:

- [1] M. Kuramoto, C. Sasaoka, N. Futagawa, M. Nido, and A. A. Yamaguchi, *physica status solidi (a)* **192**, 329 (2002).
- [2] A. A. Serkov, H. V. Snelling, S. Heusing, and T. M. Amaral, *Scientific Reports* **9**, 1773 (2019).
- [3] Y. Narukawa, J. Narita, T. Sakamoto, K. Deguchi, T. Yamada, and T. Mukai, *Japanese Journal of Applied Physics* **45**, L1084 (2006).
- [4] E. C. Young, B. P. Yonkee, F. Wu, S. H. Oh, S. P. DenBaars, S. Nakamura, and J. S. Speck, *Applied Physics Express* **9**, 022102 (2016).
- [5] B. P. Yonkee, E. C. Young, S. P. DenBaars, S. Nakamura, and J. S. Speck, *Applied Physics Letters* **109**, 191104 (2016).
- [6] M. Żak, G. Muziol, H. Turski, M. Siekacz, K. Nowakowski-Szkudlarek, A. Feduniewicz-Żmuda, M. Chlipała, A. Lachowski, and C. Skierbiszewski, *Physical Review Applied* **15**, 024046 (2021).
- [7] H. Turski, S. Bharadwaj, H. Xing, and D. Jena, *J. Appl. Phys.* **125**, 203104 (2019).
- [8] M. Siekacz *et al.*, in *Electronics2020*.
- [9] M. Siekacz *et al.*, *Opt. Express* **27**, 5784 (2019).
- [10] Y.-R. Wu, M. Singh, and J. Singh, *Journal of Applied Physics* **94**, 5826 (2003).
- [11] Y.-R. Wu and J. Singh, *Applied Physics Letters* **85**, 1223 (2004).
- [12] Y. R. Wu, C. Chiu, C. Y. Chang, P. Yu, and H. C. Kuo, *IEEE Journal of Selected Topics in Quantum Electronics* **15**, 1226 (2009).
- [13] F. Bernardini, V. Fiorentini, and D. Vanderbilt, *Physical Review B* **56**, R10024 (1997).
- [14] H. Turski *et al.*, in *Materials2022*.
- [15] S. Nakamura, M. Senoh, N. Iwasa, and S.-i. Nagahama, *Japanese Journal of Applied Physics* **34**, L797 (1995).
- [16] S. Grzanka, G. Franssen, G. Targowski, K. Krowicki, T. Suski, R. Czernecki, P. Perlin, and M. Leszczyński, *Applied Physics Letters* **90**, 103507 (2007).
- [17] M. Chlipała, H. Turski, M. Siekacz, K. Pieniak, K. Nowakowski-Szkudlarek, T. Suski, and C. Skierbiszewski, *Opt. Express* **28**, 30299 (2020).
- [18] M. A. Reshchikov, P. Ghimire, and D. O. Demchenko, *Physical Review B* **97**, 205204 (2018).
- [19] L. Eckey *et al.*, *Journal of Applied Physics* **84**, 5828 (1998).
- [20] A. David, C. A. Hurni, R. I. Aldaz, M. J. Cich, B. Ellis, K. Huang, F. M. Steranka, and M. R. Krames, *Applied Physics Letters* **105**, 231111 (2014).

REVIEWERS' COMMENTS

Reviewer #2 (Remarks to the Author):

The manuscript is well-revised. This is very nice and well-presented work.

REVIEWER COMMENTS

Reviewer #2 (Remarks to the Author):

The manuscript is well-revised. This is very nice and well-presented work.

We appreciate the insightful feedback provided by the reviewers on our manuscript titled "Bidirectional LED as an AC-driven visible-light source". The reviewers input, particularly regarding a more comprehensive description of the bidirectional LEDs operational principles and a deeper exploration of their electro-optical properties, has greatly contributed to the manuscript's improvement. After refining the manuscript according to the feedback received, we are confident that it meets the high standards of publication in Nature Communications. We thank you for your time and attention during this process.